EMBO
Molecular Medicine

# PI 3-kinase delta enhances axonal PIP₃ to support axon regeneration in the adult CNS

Bart Nieuwenhuis[1,2,†] iD, Amanda C Barber[1,†], Rachel S Evans[1,†], Craig S Pearson[1], Joachim Fuchs[3], Amy R MacQueen[4], Susan van Erp[5] iD, Barbara Haenzi[1], Lianne A Hulshof[1], Andrew Osborne[1], Raquel Conceicao[1], Tasneem Z Khatib[1], Sarita S Deshpande[1], Joshua Cave[1], Charles Ffrench-Constant[5], Patrice D Smith[6], Klaus Okkenhaug[7] iD, Britta J Eickholt[3], Keith R Martin[1,8,9], James W Fawcett[1,10] iD & Richard Eva[1,*] iD

## Abstract

Peripheral nervous system (PNS) neurons support axon regeneration into adulthood, whereas central nervous system (CNS) neurons lose regenerative ability after development. To better understand this decline whilst aiming to improve regeneration, we focused on phosphoinositide 3-kinase (PI3K) and its product phosphatidylinositol (3,4,5)-trisphosphate (PIP₃). We demonstrate that adult PNS neurons utilise two catalytic subunits of PI3K for axon regeneration: p110α and p110δ. However, in the CNS, axonal PIP₃ decreases with development at the time when axon transport declines and regenerative competence is lost. Overexpressing p110α in CNS neurons had no effect; however, expression of p110δ restored axonal PIP₃ and increased regenerative axon transport. p110δ expression enhanced CNS regeneration in both rat and human neurons and in transgenic mice, functioning in the same way as the hyperactivating H1047R mutation of p110α. Furthermore, viral delivery of p110δ promoted robust regeneration after optic nerve injury. These findings establish a deficit of axonal PIP₃ as a key reason for intrinsic regeneration failure and demonstrate that native p110δ facilitates axon regeneration by functioning in a hyperactive fashion.

**Keywords** axon transport; CNS axon regeneration; optic nerve; p110 delta; phosphoinositide 3-kinase
**Subject Categories** Neuroscience; Stem Cells & Regenerative Medicine

## Introduction

Adult central nervous system (CNS) neurons have a weak capacity for axon regeneration, meaning that injuries in the brain, spinal cord and optic nerve have devastating consequences (He & Jin, 2016; Curcio & Bradke, 2018). In most CNS neurons, regenerative capacity is lost as axons mature, both *in vitro* (Goldberg *et al*, 2002; Koseki *et al*, 2017) and *in vivo* (Kalil & Reh, 1979; Wu *et al*, 2007). Conversely, peripheral nervous system (PNS) neurons maintain regenerative potential through adult life. This is partly because PNS neurons mount an injury response in the cell body (Smith & Skene, 1997; Ylera *et al*, 2009; Puttagunta *et al*, 2014), and also because PNS axons support efficient transport of growth-promoting receptors, whilst many of these are selectively excluded from mature CNS axons (Hollis *et al*, 2009a,b; Franssen *et al*, 2015; Andrews *et al*, 2016). Studies into intrinsic regenerative capacity have implicated signalling molecules, genetic factors and axon transport pathways as critical regeneration determinants (Park *et al*, 2008; Blackmore *et al*, 2012; Fagoe *et al*, 2015; Eva *et al*, 2017; Weng *et al*, 2018; Hervera, 2019). This leads to a model where axon growth capacity is controlled by genetic and signalling events in the cell body, and by the selective transport of growth machinery into the axon to re-establish a growth cone after injury.

To better understand the mechanisms regulating axon regeneration, we focused on the class I phosphoinositide 3-kinases (PI3Ks). These enzymes mediate signalling through integrins, growth factor and cytokine receptors, by producing the membrane phospholipid PIP₃ from PIP₂ (phosphatidylinositol (3,4,5)-trisphosphate from

1   John Van Geest Centre for Brain Repair, Department of Clinical Neurosciences, University of Cambridge, Cambridge, UK
2   Laboratory for Regeneration of Sensorimotor Systems, Netherlands Institute for Neuroscience, Royal Netherlands Academy of Arts and Sciences (KNAW), Amsterdam, The Netherlands
3   Institute of Biochemistry, Charité − Universitätsmedizin Berlin, Berlin, Germany
4   Laboratory of Lymphocyte Signalling and Development, Babraham Institute, Cambridge, UK
5   MRC Centre for Regenerative Medicine, University of Edinburgh, Edinburgh, UK
6   Department of Neuroscience, Carleton University, Ottawa, ON, Canada
7   Department of Pathology, University of Cambridge, Cambridge, UK
8   Centre for Eye Research Australia, Royal Victorian Eye and Ear Hospital, Melbourne, Vic., Australia
9   Ophthalmology, Department of Surgery, University of Melbourne, Melbourne, Vic., Australia
10  Centre of Reconstructive Neuroscience, Institute of Experimental Medicine, Czech Academy of Sciences, Prague, Czech Republic
    *Corresponding author. Tel: +44 1223 331188; E-mail: re263@cam.ac.uk
    †These authors contributed equally to this work

phosphatidylinositol(4,5)-bisphosphate). Class 1 PI3Ks comprise 4 catalytic isoforms called p110α, β, γ and δ, with distinct roles for some of these emerging in specific cell populations (Bilanges *et al*, 2019). The p110α and p110β isoforms are ubiquitously expressed, whilst p110δ and p110γ are highly enriched in leucocytes (Hawkins & Stephens, 2015). p110β and p110γ have not been studied in neurons, but p110α mediates axon growth during chick development (Hu *et al*, 2013) whilst p110δ is required for axon regeneration of the PNS (Eickholt *et al*, 2007) and localises to the Golgi in mature cortical neurons, where it controls the trafficking of the amyloid precursor protein (APP) (Low *et al*, 2014; Martinez-Marmol *et al*, 2019).

The PI3K pathway is strongly implicated in the regulation of regenerative ability because transgenic deletion of PTEN promotes CNS regeneration (Park *et al*, 2008; Liu *et al*, 2010; Geoffroy *et al*, 2015). PTEN opposes PI3K by converting PIP$_3$ back to PIP$_2$. Additionally, inhibiting negative feedback of this pathway similarly enhances regrowth (Al-Ali *et al*, 2017). These findings indicate a pro-regenerative role for PIP$_3$; however, this molecule has not been directly studied in adult CNS axons. Axonal PI3K contributes to polarity in developing hippocampal axons (Shi *et al*, 2003), and PNS axons segregate PI3K at the growth cone to elicit rapid growth (Zhou *et al*, 2004); however, the neuronal distribution of PIP$_3$ is not known. We reasoned that CNS regenerative failure might be associated with a developmental decline in PIP$_3$ specifically within the axon, and wondered whether individual PI3K isoforms were required to yield sufficient axonal PIP$_3$.

We investigated the class I PI3K isoforms and found that both p110α and p110δ are required for PNS axon regeneration and that p110δ is specifically required within the axon. In CNS neurons, we found PIP$_3$ was sharply downregulated with development, diminishing in the axon at the time when axon transport and regeneration also decline. We attempted to restore PIP$_3$ through overexpression of either p110α or p110δ; however, only p110δ led to elevated axonal PIP$_3$. Importantly, by introducing the hyperactivating H1047R mutation into p110α we found that it could mimic the effect of p110δ, with the expression of either p110δ or p110α$^{H1047R}$ facilitating axon regeneration. This suggests that regeneration is hindered by low activation of PI3K in mature CNS axons. We found that transgenic expression of p110δ or p110α$^{H1047R}$ led to enhanced retinal ganglion cell (RGC) survival and axon regeneration after optic nerve crush, whilst viral expression of p110δ led to stronger regeneration. Importantly, overexpression of p110δ had both somatic and axonal effects, enhancing the axonal transport of growth machinery and signalling through ribosomal S6 in the cell body. These findings demonstrate a deficit of axonal PIP$_3$ as a novel reason for intrinsic regenerative failure, whilst establishing that native p110δ functions in a hyperactive fashion to enable CNS axon regeneration. Our results emphasise the importance of elevating growth-promoting pathways in the axon as well as the cell body to stimulate axon regeneration.

# Results

## Gene expression of p110 isoforms in the nervous system

Because localised axonal activation of PI3K is essential for rapid PNS axon growth (Zhou *et al*, 2004), we reasoned that there may

be specific PI3K isoforms that support regeneration within PNS axons and that these might be under-represented in CNS neurons. p110δ plays a role in PNS axon regeneration (Eickholt *et al*, 2007); however, the contribution of other isoforms has not been examined. We investigated the RNA expression of the individual p110 subunits (p110α, β, γ and δ) in four published neuronal RNAseq datasets (Fig EV1), examining expression in dorsal root ganglion (DRG) neurons during axon growth and regeneration (Tedeschi *et al*, 2016), in developing cortical neurons (Koseki *et al*, 2017) and in the Brain-RNAseq databases (http://www.brainrnaseq.org/) of individual cell types in mouse and human brain (Zhang *et al*, 2014, 2016). In DRG neurons, p110α, β and δ are expressed at all stages *in vitro*, whilst p110γ is expressed at very low levels. p110α is present at the highest levels, whilst p110δ is upregulated in development and further upregulated upon peripheral nerve lesion (Fig EV1A–C). In developing cortical neurons, p110α is again expressed at the highest levels, p110β and δ are present but not abundant, and p110γ is at very low levels. Remarkably, p110α, β and δ are all downregulated as cortical neurons mature (Fig EV1D and E). The Brain-RNAseq database indicates p110α and β are the principal neuronal PI3K isoforms in both the mouse and human adult brain, with p110α again expressed at the highest levels, whilst p110δ and γ are enriched in microglia and macrophages (Fig EV1F–M). These datasets indicate that p110α, β and δ are expressed in regenerative PNS neurons, but are almost absent or downregulated with maturation in CNS neurons. We therefore chose to investigate the contribution of the p110α, β and δ isoforms of PI3K to axon regeneration of DRG neurons.

## p110α and δ are required for DRG axon regeneration

We used laser axotomy to sever the axons of adult DRG neurons and measured the effect of specific inhibitors of p110α, β and δ on growth cone regeneration (Fig 1). Inhibiting either p110α or δ reduced the percentage of axons regenerating, as did pan-PI3K inhibition (α, β and δ) or targeting p110α and δ together. Inhibition of p110β had no effect, but inhibiting all of the isoforms increased the time taken to develop a new growth cone (Fig 1A and B and Movies EV1 and EV2). Some inhibitors also caused uncut axons to stop growing, so we measured the extension rate of uncut axons. Treatment with p110α inhibitors, pan-PI3K or dual α/δ inhibitors led to a dramatic reduction in the percentage of uncut axons extending in 2 h, whilst they continued to extend in the presence of specific p110δ inhibitors (Fig 1C and D and Movies EV3 and EV4). We next examined the effect of PI3K inhibition using microfluidic compartmentalised chambers, in which axons extend through microchannels into a separate compartment from the cell bodies (Fig 1E). Inhibition of p110α or p110δ showed different effects depending on localisation. Inhibition of p110δ in the axonal compartment reduced the percentage of regenerating axons, but had no effect in the somatic chamber. In contrast, the p110α inhibitor A66 reduced regeneration when applied either to the axonal or somatic compartment, and also increased the time taken to generate a new growth cone (Fig 1F). These data show that DRG neurons require p110α and δ for efficient regeneration. Axon growth and regeneration require p110α activity in both the cell body and axon, whilst axon regeneration further relies on p110δ activity specifically within the axon.

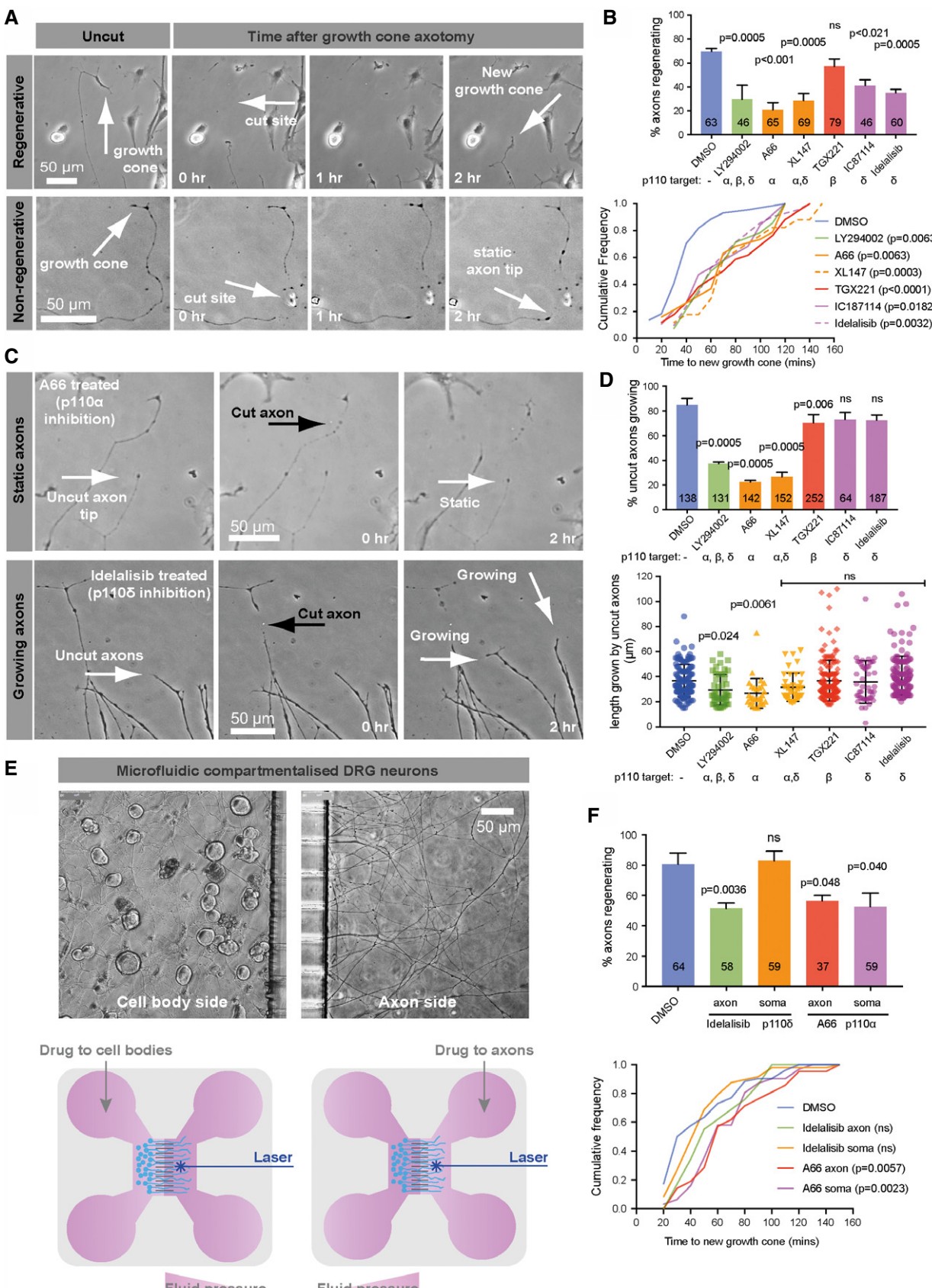

Figure 1.

**Figure 1.  p110α and δ are required for rat DRG axon regeneration; p110δ functions in the axonal compartment.**

A   Laser-injured DRG axons showing growth cone regeneration (upper panels) and regenerative failure (lower panels). Arrows as indicated.
B   Upper graph shows percentage of regenerating axons of DIV 1–2 adult DRG neurons in the presence of p110 inhibitors, 2 h after injury. Lower graph shows time taken to regenerate a new growth cone. Inhibitors were added 1 h before axons were injured and maintained throughout: LY294002, 20 µM; A66, 5 µM; XL-147, 5 µM; TGX221, 500 nM; IC-87114, 10 µM; idelalisib, 500 nM. Upper graph data are shown as the mean ± SEM. *P*-values indicate statistical significance analysed by Fisher's exact (upper graph) or Kruskal–Wallis test (lower graph).
C   Examples of static vs. growing uninjured axons treated with p110α or δ inhibitors as indicated. Both arrows in the bottom right image indicate growing axons.
D   Percentage of uninjured, growing axons in the presence of p110 inhibitors. Inhibitors were added as for (B). Lower graph shows length grown in 2 h. Data are shown as the mean ± SEM. *P*-values indicate significance measured by Fisher's exact test (upper graph) or ANOVA with Tukey's *post-hoc* analysis (lower graph).
E   Adult DRG neurons in microfluidic compartmental chambers. Cell bodies on the left side, axons extending through microchannels on the right. Lower panels are schematics.
F   Upper graph shows percentage of regenerating axons in microfluidic chambers. p110α or δ inhibitors were applied to either soma or axons. Lower graph shows time taken to regenerate a new growth cone. Inhibitors were added 1hr before axons were injured and maintained throughout: A66, 5 µM; idelalisib, 500 nM. Upper graph data are shown as the mean ± SEM. *P*-values indicate statistical significance as measured by Fisher's exact test (upper graph) or by Kruskal–Wallis test (lower graph).

Data information: Numbers on bars are the total numbers of axons cut, from n = 4 experiments for (B, D and F).

## PIP₃ is developmentally downregulated in cortical neurons, but present in adult DRG axons

The RNAseq data described above (Fig EV1) suggest that PI3K expression increases with maturation in DRG neurons, whilst it is downregulated as cortical neurons mature. CNS regenerative failure might therefore be due to insufficient PI3K activity within the axon and insufficient $PIP_3$. $PIP_3$ is implicated in axon growth but its developmental distribution has not been examined. $PIP_3$ has previously been localised using fluorescently tagged pleckstrin homology (PH)-domain reporters, such as AKT-PH-GFP; however, the principle readout of these reporters is translocation to the surface membrane, and they do not report on abundance or "steady-state" distribution. To accurately measure $PIP_3$ in neurons developing *in vitro*, we optimised a fixation technique (Hammond *et al*, 2009) for antibody-based $PIP_3$ detection on immobilised membrane phospholipids utilising an antibody widely used for biochemical assays. To validate this in neurons, we isolated DRG neurons from transgenic mice expressing AKT-PH-GFP at low levels (Nishio *et al*, 2007) to avoid inhibition of downstream signalling (Varnai *et al*, 2005). Live imaging of adult DRG neurons from these mice revealed dynamic hotpots of $PIP_3$ at the axon growth cone (Fig EV2A and Movie EV5), and membrane labelling confirmed these are not regions of membrane enrichment (Fig EV2B and Movie EV6). Phospholipid fixation and labelling with anti-$PIP_3$ revealed colocalisation between AKT-PH-GFP and anti-$PIP_3$ at regions within DRG growth cones (Fig EV2C), as well as at hotspots and signalling platforms in non-neuronal cells from DRG cultures (Fig EV2D). In addition to validating this technique, our data also confirm the presence of dynamic $PIP_3$ in the growth cone and axons of regenerative DRG neurons. To further confirm the specificity of the stain for $PIP_3$, we stimulated N1E cells with insulin and labelled for $PIP_3$ in the presence or absence of the pan-PI3K inhibitor GDC-0941, detecting increased $PIP_3$ staining after insulin stimulation alone, and not in the presence of the PI3K inhibitor (Fig EV2E–G).

We then labelled E18 cortical neurons at 3, 8 and 16 days *in vitro* (DIV) to detect endogenous $PIP_3$. In immature neurons (at DIV 3), we detected high levels of $PIP_3$ in the cell body, and particularly in the distal axon and growth cone (Fig 2A, D and E). At DIV 8, these neurons have a single rapidly growing axon; at this time, there was a sharp decline in $PIP_3$ at the cell body (Fig 2B and D), but only a small decrease in growth cone $PIP_3$ (Fig 2B and E), indicating that during the period of rapid axon extension, neurons possess high levels of $PIP_3$ in axonal growth cones. By DIV 16, cortical neurons have long axons, which are establishing synapses, with some branches still growing. These mature neurons exhibit a sharp decline in regenerative ability and selective axon transport (Eva *et al*, 2017; Koseki *et al*, 2017). At this stage, we detected markedly reduced levels of $PIP_3$ at the growth cone, whilst they remained low at the cell body (Fig 2C–E). This indicates a global reduction in $PIP_3$ as cortical neurons develop and demonstrates that the loss of regenerative ability coincides with a deficiency in axonal $PIP_3$ production.

## Expression of p110δ or p110α^H1047R elevates PIP₃ in the soma and axon

The results above show that PI3K and $PIP_3$ are developmentally downregulated in CNS axons as they lose their regenerative ability. In DRG neurons, which continue to express PI3K and produce $PIP_3$, the p110α and δ isoforms are necessary for efficient axon regeneration (Fig 1). We therefore asked whether we could increase $PIP_3$ levels and restore regeneration to CNS neurons by ectopic

**Figure 2.  PIP₃ is developmentally downregulated in rat cortical neurons maturing *in vitro*, first in the soma, then at the growth cone.**

A   DIV 3 cortical neurons immunolabelled for $PIP_3$. Spectrum heatmap shows fluorescence intensity.
B   DIV 8 cortical neuron transfected with GFP and immunolabelled for $PIP_3$. Spectrum heatmap shows fluorescence intensity.
C   DIV 16 cortical neuron transfected GFP and immunolabelled for $PIP_3$. Spectrum heatmap shows fluorescence intensity.
D   Somatic $PIP_3$ quantification in the soma at increasing days *in vitro*. Data are shown as the mean ± SEM. *P*-values are significance measured by ANOVA with Tukey's *post-hoc* analysis. n = 3 experiments, 60 neurons.
E   Growth cone $PIP_3$ quantification at increasing days *in vitro*. Data are shown as the mean ± SEM. *P*-values indicate significance measured by ANOVA with Tukey's *post-hoc* analysis. n = 3 experiments, 60 growth cones.

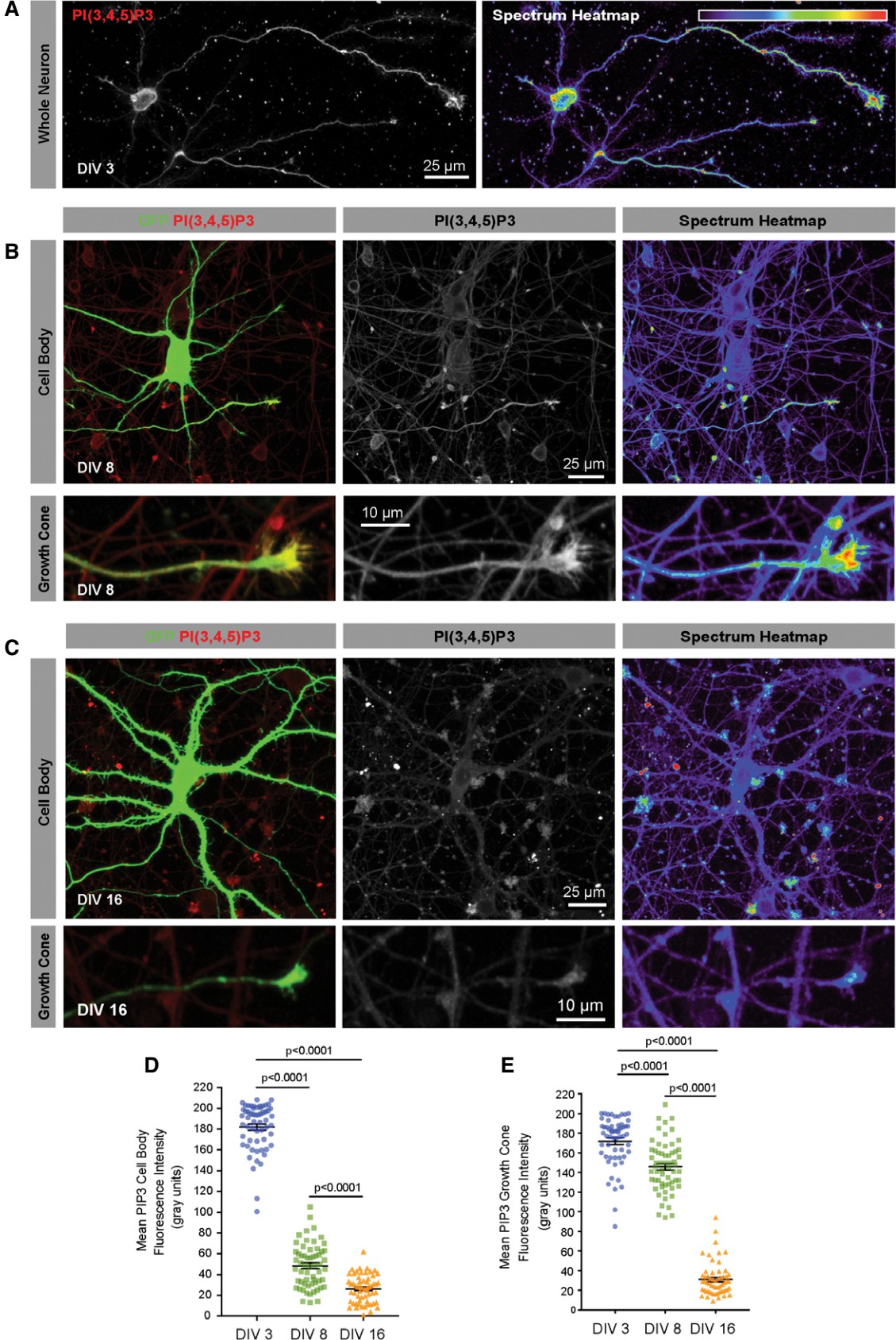

**Figure 2.**

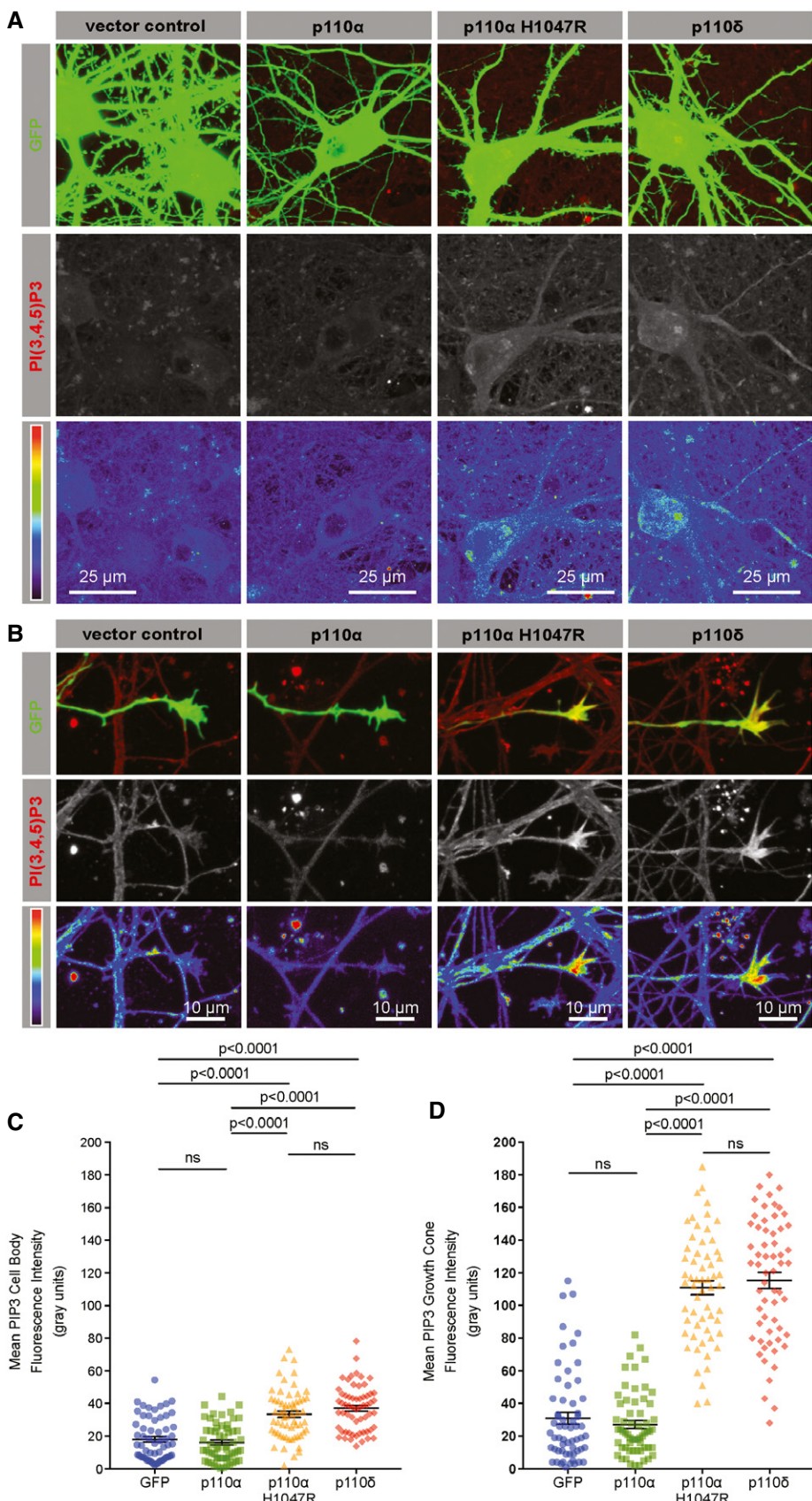

Figure 3.

◀

**Figure 3.  p110δ or p110α^H1047R expression elevates PIP3 in the soma and axon, whilst native p110α does not.**

A   PIP3 immunofluorescence in the soma of DIV 16 cortical neurons expressing p110 isoforms and GFP. Spectrum heatmap shows fluorescence intensity.
B   PIP3 immunofluorescence at the axonal growth cone of DIV 16 neurons expressing p110 isoforms and GFP. Spectrum heatmap shows fluorescence intensity.
C   Quantification of PIP3 immunofluorescence in the soma. Data are shown as the mean ± SEM. *P*-values indicate significance as measured by ANOVA with Tukey's *post-hoc* analysis. n = 3 experiments, 60 neurons.
D   Graph showing PIP3 quantification of immunofluorescence at the axon growth cone. Data are shown as the mean ± SEM. *P*-values indicate significance measured by ANOVA with Tukey's *post-hoc* analysis. n = 3 experiments, 60 growth cones.

expression of p110α, its hyperactive H1047R variant p110α^H1047R (Mandelker *et al*, 2009) or p110δ. Previous work has shown that p110δ and p110α^H1047R can sustain downstream AKT activation upon overexpression in fibroblasts, acting independently of active Ras, whilst native p110α does not (Kang *et al*, 2006). These PI3Ks were expressed in cortical neurons at DIV 16, and PIP3 was measured by quantitative immunofluorescence. We used dual-promoter constructs expressing untagged p110 and GFP to avoid potential interference of PI3K function by a protein tag.

Compared with cells expressing GFP alone, expression of p110α had no effect on PIP3 levels in DIV 16 neurons, either in the cell body or within the axons (Fig 3A–D). In contrast, overexpression of either p110α^H1047R or p110δ led to a small but significant increase in PIP3 in the cell body (Fig 3A and C) and a striking increase in PIP3 at axon growth cones (Fig 3B and D). These findings indicate that p110δ and p110α^H1047R function in a hyperactive fashion to generate PIP3 in cortical neurons, with the strongest effect in growth cones. However, even overexpressed p110α does not generate PIP3, suggesting a low level of activation in mature neurons.

**Expression of p110δ or p110α^H1047R increases axon and dendrite growth**

We next investigated the effect of p110δ, p110α or p110α^H1047R overexpression on the regulation of axon growth in cortical neurons developing *in vitro*. Expression of either p110δ or p110α^H1047R in immature neurons at DIV 2 led to a moderate increase in axon length by DIV 4, compared with neurons expressing either GFP alone or native p110α (Fig 4A and B). p110δ or p110α^H1047R expression also led to a small increase in the dendrite length (Fig 4A and C). This led to an increase in axon/dendrite length ratio, with axons approximately 31.5 times longer than dendrites, whilst control neurons have axons 21.8 times longer than dendrites. None of the PI3K isoforms therefore affected neuronal polarisation (Fig 4A and D). We also examined the effects of PI3K overexpression on dendrite length and branching at a later developmental stage (transfecting at DIV 10, and analysing at DIV 14). Again, p110δ and p110α^H1047R behaved similarly, expression of either construct leading to increases in both the number of dendrite branches and the total dendrite length, compared with control-transfected neurons. In contrast, expression of native p110α had no effect (Fig 4E–G). The PI3K pathway is a well-known regulator of cell size, so we also measured hypertrophy. Overexpression of either p110δ or p110α^H1047R led to an increase in cell body size, compared with either GFP or p110α (Fig 4H). To confirm downstream signalling through the PI3K pathway, we employed phosphorylation-specific immunolabelling of ribosomal S6 protein, a transcriptional regulator routinely used as a reporter of somatic

signalling through the PI3K/AKT/mTOR pathway. p110δ or p110α^H1047R expression led to a strong phospho-S6 signal compared with GFP-expressing controls, whilst expression of p110α again had no effect (Fig 4E and I). These findings confirm that hyperactive p110α^H1047R behaves like p110δ to trigger downstream signalling through the PI3K pathway in neuronal soma, with effects on size, dendrite branching and axon length. The results demonstrate that p110δ and p110α^H1047R enhance both axonal and dendritic growth, whilst native p110α does not.

**p110δ and p110α^H1047R promote axon regeneration of CNS neurons *in vitro***

To determine whether activation of p110 can facilitate CNS regeneration, we used an *in vitro* model of regeneration in mature cortical neurons, comparing overexpression of either p110δ, p110α or p110α^H1047R. In this model, axon regeneration ability is progressively lost with maturity, and molecules that promote regeneration may differ from those that promote developmental outgrowth (Koseki *et al*, 2017). *In vitro* laser axotomy was used to sever the axons of E18 cortical neurons cultured to DIV 15–18, at which stage they have a limited capacity for regeneration (Eva *et al*, 2017; Koseki *et al*, 2017). Expression of either p110δ or p110α^H1047R led to a sharp increase in the percentage of axons regenerating after axotomy compared with GFP-expressing controls. p110α expression again had no effect (Fig 5A and B and Movies EV7 and EV8). Expression of p110δ or p110α^H1047R also led to an increase in the length of regenerated axons and a trend towards shorter time of onset to regeneration compared to controls (Fig 5C and D).

We further investigated the translational potential of p110δ by examining axon regeneration in human neurons maturing *in vitro*, derived from human embryonic stem cells (hESC). We have previously demonstrated that these lose their regenerative ability *in vitro* when cultured beyond 50 days (Koseki *et al*, 2017). Overexpression of p110δ fully restored the regenerative ability to a level observed for younger neurons (Fig 5E and F). We observed no difference in the length of regenerated axons compared with GFP-expressing control neurons (Fig 5G), but there was a tendency for p110δ expressing neurons to initiate regeneration faster (Fig 5H). Overexpression of p110δ therefore enables CNS axon regeneration in both rat and human neurons, and this is mimicked by hyperactive p110α^H1047R, whilst overexpression of p110α is ineffective.

**p110δ functions through multiple downstream pathways**

The data presented so far demonstrate that p110δ expression elevates axonal PIP3 and enhances axon regeneration. We performed a series of experiments to investigate the downstream pathways involved, focusing on mTOR, CRMP2 and ARF6. Deletion

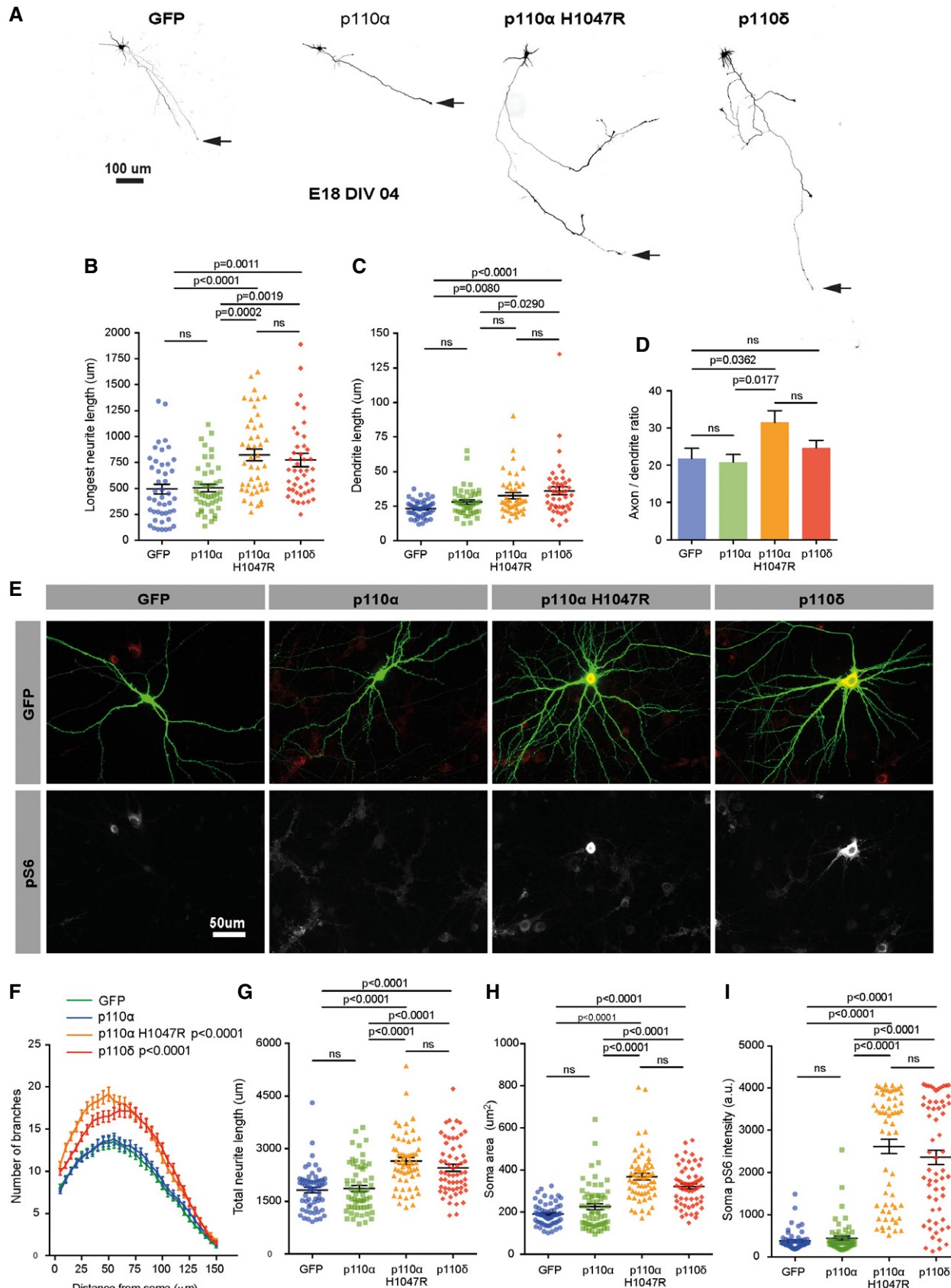

**Figure 4.**

**Figure 4. Expression of p110δ or p110α^H1047R increases axon and dendrite growth of rat cortical neurons developing *in vitro*, whilst native p110α does not.**

- A   DIV 4 cortical neurons expressing p110 isoforms and GFP. Arrow marks the axon tip.
- B   Quantification of axon length. n = 3 experiments, 45 neurons per group.
- C   Quantification of dendrite length. n = 3 experiments, 45 neurons per group.
- D   Quantification of the axon:dendrite length ratio. n = 3 experiments, 45 neurons per group.
- E   DIV 14 cortical neurons expressing p110 isoforms and GFP, immunolabelled for pS6.
- F   Sholl analysis of branches. n = 60 neurons per group. Data are shown as the mean ± SEM. *P*-values indicate significance measured by repeated-measure ANOVA with Bonferonni's *post-hoc* test.
- G   Quantification of the total neurite length. n = 3 experiments, 60 neurons per group.
- H   Quantification of soma area. n = 3 experiments, 60 neurons per group.
- I   Quantification of the pS6 immunofluorescence. n = 3 experiments, 60 neurons per group.

Data information: (B–D and G–I) Data are shown as the mean ± SEM. *P*-values indicate significance measured by ANOVA with Tukey's *post-hoc* analysis.

of PTEN and PI3K activity promotes regeneration by signalling through mTOR (Park *et al*, 2008) and through GSK3 and CRMP2 (Leibinger *et al*, 2019). Deletion of PTEN or elevation of PI3K activity could also affect signalling through many molecules with PH, PX or FYVE domains (Vanhaesebroeck *et al*, 2012). One molecule of particular interest is the small GTPase ARF6, which regulates regenerative ability in both PNS and CNS neurons by controlling the axonal supply of integrins in Rab11-positive endosomes, governed by its activation state (Eva *et al*, 2017). In CNS neurons, axonal ARF6 activation is maintained by two ARF6 GEFs, ARNO and EFA6, which results in predominant retrograde transport away from the axon (Franssen *et al*, 2015; Eva *et al*, 2017). EFA6 and ARNO interact in a complicated fashion to sustain active ARF6, regulated by their PH domains (Malaby *et al*, 2013; Padovani *et al*, 2014). We reasoned that elevated PI3K activity might be associated with a decrease in axonal ARF6 activation, and an increase in the anterograde transport of integrins in Rab11 endosomes.

We used laser axotomy and pharmacological inhibitors on mature cortical neurons to test the contribution of mTOR and CRMP2 to the regenerative effects of p110δ, and to confirm that p110δ enabled regeneration through its kinase activity. Overexpression of p110δ again led to a robust increase in the percentage of axons regenerating after laser injury; however, this was prevented by the presence of the specific p110δ inhibitor idelalisib, whilst the p110α inhibitor A66 had no effect (Fig 6A). To investigate the role of mTOR and CRMP2 in p110δ-mediated axon regeneration, we applied rapamycin, an inhibitor of mTOR, or lacosamide, a specific inhibitor of CRMP2, which has been previously used to study CRMP2 and regenerative growth *in vitro* (Leibinger *et al*, 2019). Both drugs inhibited the regenerative effects of p110δ expression to similar degrees (Fig 6A), confirming that both pathways are functioning downstream of p110δ activity to mediate regeneration. The increase in phospho-S6 observed in p110δ expressing neurons (Fig 4E) further supports the involvement of the mTOR pathway.

We also examined the localisation of overexpressed p110δ. Endogenous p110δ has been observed at the Golgi, where it controls the trafficking of cytokines and APP (Low *et al*, 2010, 2014; Martinez-Marmol *et al*, 2019). We did not find overexpressed p110δ enriched at any specific subcellular structures, but instead found it distributed throughout the neuron (Fig 6B) in the somatodendritic domain (upper panel), axon (middle panel) and axon growth cone (lower panels), whilst being excluded from the nucleus. Overexpressed p110δ may therefore exert its effects throughout neurons, including at the Golgi, with the exception of the nucleus.

To investigate ARF6 activation, we visualised activated ARF6 in the axons of DIV 16 neurons by using the ARF-binding domain

(ABD) of GGA3 fused to a GST tag, which was previously used to measure ARF6 activation in neurons (Eva *et al*, 2017). Overexpression of p110δ led to a reduction in ARF6 activation compared to control neurons expressing GFP alone, whilst we detected no change in the amount of total ARF6 protein (Fig 6C and D). We next investigated whether the reduction of ARF6 activation was accompanied by an increase in integrin or Rab11 transport. ARF6 controls directional transport in a complex with Rab11 (Montagnac *et al*, 2009), a marker of recycling endosomes that controls neuronal integrin traffic, as well as other growth receptors (Ascano *et al*, 2009; Eva *et al*, 2010; Lazo *et al*, 2013), and contributes to the regenerative ability of cortical neurons (Koseki *et al*, 2017). We used spinning disc microscopy to image integrin dynamics in the distal axon by visualising α9-integrin–GFP, in the presence of mCherry (control) or mCherry plus p110δ. The presence of mCherry allows visualisation of the entire axon. Anterograde transport was almost undetectable in control-transfected neurons where we observed predominantly retrograde and static vesicles (Fig 6E and F), in keeping with previous studies (Franssen *et al*, 2015). Expression of p110δ triggered anterograde movement of integrins, leading to an increase in static integrin vesicles in the distal axon, and an overall increase in the total number of integrin vesicles (Fig 6G). We also analysed the dynamics of Rab11-GFP vesicles. These moved throughout axons in an oscillatory fashion displaying mostly bidirectional and retrograde movements, in keeping with previous findings (Eva *et al*, 2017); however, expression of p110δ initiated anterograde transport and caused an increase in static and bidirectional movements. This led to an overall increase in Rab11-GFP vesicles in the distal part of p110δ expressing axons, compared with controls (Fig 6H–J). Expression of p110δ therefore enhances the anterograde transport of alpha9 integrins and Rab11-positive endosomes. These transport changes may occur as a result of a reduction in axonal ARF6 activation, although it is also possible that signalling through CRMP2 may also contribute (Rahajeng *et al*, 2010). Additionally, p110δ may also be functioning at the Golgi to regulate anterograde transport, as has been shown previously (Martinez-Marmol *et al*, 2019). Together, our findings demonstrate that p110δ functions to support regeneration through multiple downstream pathways, including the increased transport of regenerative machinery.

**Transgenic p110δ and p110α^H1047R support RGC survival and axon regeneration in the optic nerve**

To confirm whether p110δ and p110α^H1047R support regeneration in the adult CNS *in vivo*, as we found *in vitro*, we used the optic nerve

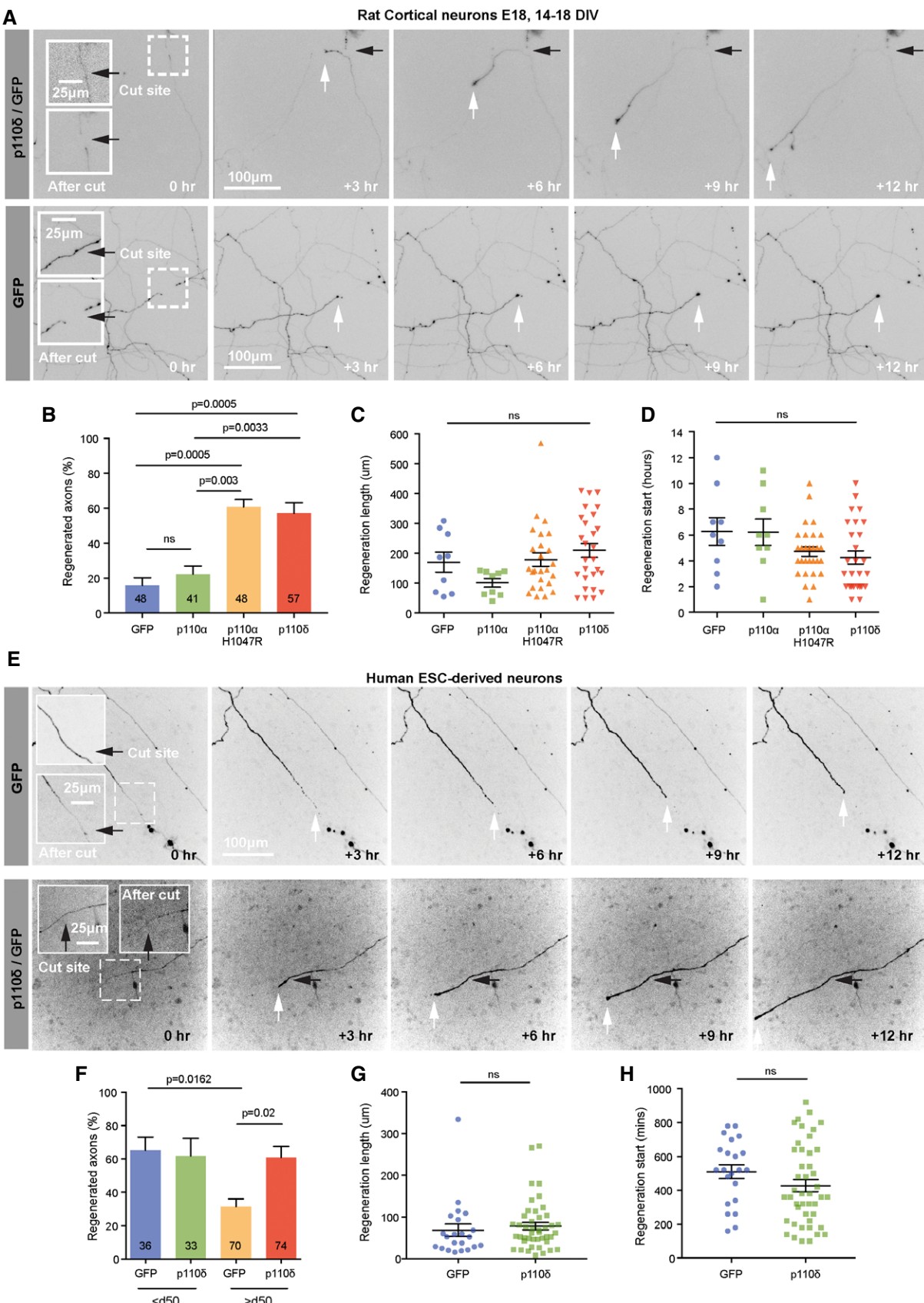

Figure 5.

**Figure 5.  p110δ and p110α^H1047R promote axon regeneration of CNS neurons *in vitro*.**

A   Axotomised DIV 14–16 rat cortical neurons expressing p110δ and GFP or GFP alone. Axons were cut > 1,000 μm from the soma imaged for 14 h. Black arrows mark the cut site, and white arrows mark the axon tip. See also Movies EV7 and EV8.

B   Percentage of regenerating axons 14 h after laser axotomy. Numbers on bars are injured axons per group, from 4 experiments.

C   Quantification of axon regeneration length 14 h post-axotomy. n = 9 axons for GFP, 9 for p110α, 25 for p110α^H1047R and 27 for p110δ from 4 experiments.

D   Quantification of time to start of regeneration. n = 9 axons for GFP, 9 for p110α, 25 for p110α^H1047R and 27 for p110δ from 4 experiments.

E   Axotomised human embryonic stem cell neurons p110δ and GFP or GFP alone. Black arrow marks the cut site, and white arrows mark the axon tip.

F   Percentage of regenerating axons of human hESC neurons. Numbers on bars are injured axons per group, from 5 experiments.

G   Quantification of axon regeneration length. n = 22 axons for GFP, 44 for p110δ, from 5 experiments. Data are shown as the mean ± SEM. Data were analysed by two-tailed Student's *t*-test.

H   Quantification of time to start of regeneration. n = 22 axons for GFP, 44 for p110δ, from 5 experiments. Data are shown as the mean ± SEM. Data were analysed by two-tailed Student's *t*-test.

Data information: (B–D and F), Data are shown as the mean ± SEM. *P*-values indicate significance measured by ANOVA with Tukey's *post-hoc* analysis.

crush model to examine the effects of PI3K activation on regeneration after a crush injury. We used transgenic mice, which conditionally express either p110δ or p110α^H1047R from the Rosa26 locus in the presence of Cre recombinase (Fig 7A), and delivered AAV2-Cre-GFP via intravitreal injection. Rosa26 allows expression of a transgene at moderate levels (Nyabi *et al*, 2009). Two weeks after viral injection, optic nerve crush was performed, and retinal ganglion cell (RGC) survival and optic nerve regeneration were examined 28 days later (Fig 7B). We tested AAV2-Cre-GFP activity using a Cre-reporter mouse, which expresses tdTomato from the Rosa26 locus (Fig EV3A) and confirmed expression in p110δ and p110α^H1047R mice by examining GFP expression (from the viral vector) in the retina (Fig EV3B). Activation downstream of PI3K was confirmed by phospho-S6 immunofluorescence. Expression of either p110δ or p110α^H1047R led to an increase in the number of cells labelling positive for phospho-S6; however, p110α^H1047R expression led to a slightly larger increase than p110δ expression (Fig 7C and D). p110δ and p110α^H1047R behaved similarly with respect to their effects on RGC survival. The presence of transgenic p110δ or p110α^H1047R led to a doubling of the number of cells surviving after 28 days (from 5.5 to 11%), demonstrating a strong neuroprotective effect (Fig 7E and F). We then measured axon regeneration in the optic nerve and found that p110α^H1047R and p110δ again behaved similarly, both enabling a moderate increase in axon regeneration compared with control mice injected with AAV2-Cre.GFP (Fig 7G and H). The results confirm that p110δ and p110α^H1047R behave similarly in injured RGC neurons in the CNS *in vivo*, enhancing RGC survival and axon regeneration.

## AAV2-p110δ facilitates axon regeneration in the optic nerve

The moderate effects of transgenic p110 expression on axon regeneration (described above) were surprising, given the robust effects of p110δ or p110α^H1047R expression on CNS axon regeneration *in vitro*. We reasoned that p110 enabled axon regeneration may be dose-dependent and that the moderate expression generated from the Rosa26 locus (Nyabi *et al*, 2009) might explain these limited effects. To test whether regeneration could be enhanced using a viral gene transfer approach, we produced an AAV2-p110δ construct for viral transduction of RGC neurons via intravitreal injection. We compared this with a similar AAV2-mediated shRNA approach to silence PTEN, the phosphatase responsible for opposing the actions of PI3K by dephosphorylating PIP$_3$ to PIP$_2$. Transgenic suppression of PTEN is another means of stimulating

regeneration in the CNS, although virus-mediated shRNA silencing has not proved to be as effective as transgenic deletion (Yungher *et al*, 2015). We therefore compared viral vector-based delivery of p110δ *versus* viral delivery of a PTEN targeting shRNA. We first confirmed that AAV2-shPTEN-GFP transduction resulted in silencing of PTEN in RGCs compared with AAV2-scrambled-GFP control, PTEN levels being measured by quantitative immunofluorescence (Fig EV4A and B). We confirmed transduction of RGCs by AAV2-p110δ immunofluorescence, comparing p110δ with RGC neurons transduced with AAV2-GFP (Fig EV4C and D). We examined downstream signalling by labelling for ribosomal phospho-S6. Transduction with AAV2-shPTEN-GFP led to an increase in the percentage of transduced cells labelling positive for phospho-S6 compared with AAV2-scrambled-GFP-transduced neurons (from 27% for controls to 48% for PTEN silenced; Fig 8A and B). Due to the lack of a tag on AAV2-p110δ (due to potential effects on activity), we measured the total number of TUJ1 labelled neurons that also labelled for phospho-S6. Transduction with AAV2-p110δ led to an increase in phospho-S6-positive neurons compared with AAV2-GFP-transduced controls (1140 cells for p110δ transduced, compared with 650 cells for GFP transduced) (Fig 8A and C).

Having validated the two approaches, we next compared their effects on RGC survival and axon regeneration after optic nerve crush. Silencing of PTEN led to a robust increase in the survival of RGC neurons compared with controls at 28 days after crush (14.7% of PTEN-silenced neurons survived compared to 4.9% of control neurons), whilst transduction with p110δ led to a smaller effect on survival (11.3% of p110δ-transduced neurons compared 6.3% of GFP control neurons; Fig 8D and E), similar to the amount observed due to transgenic expression (Fig 7F). We examined CTB-traced RGC axon regeneration in the optic nerve and found that both AAV2-mediated silencing of PTEN and AAV2 delivery of p110δ enhanced axon regeneration. Transduction with p110δ had the most robust effects on regeneration, with 180 axons counted at 0.5 mm into the optic nerve, compared with 25 axons in control-injected mice (Fig 8F). Injection of AAV2-shPTEN-GFP led to 97 axons at 0.5 mm, again compared with 25 axons for controls (this compares with 60 axons for p110δ, 70 for p110α^H1047R with transgenic expression). AAV2-p110δ also enabled axons to regenerate further into the optic nerve reaching a maximum distance of 3 mm, whilst silencing of PTEN enabled axons to reach a maximum of 1.5 mm (transgenic expression of p110α^H1047R and p110δ also gave 1.5 mm regeneration). These data demonstrate that the PI3K pathway can be targeted to stimulate RGC survival and axon regeneration either by

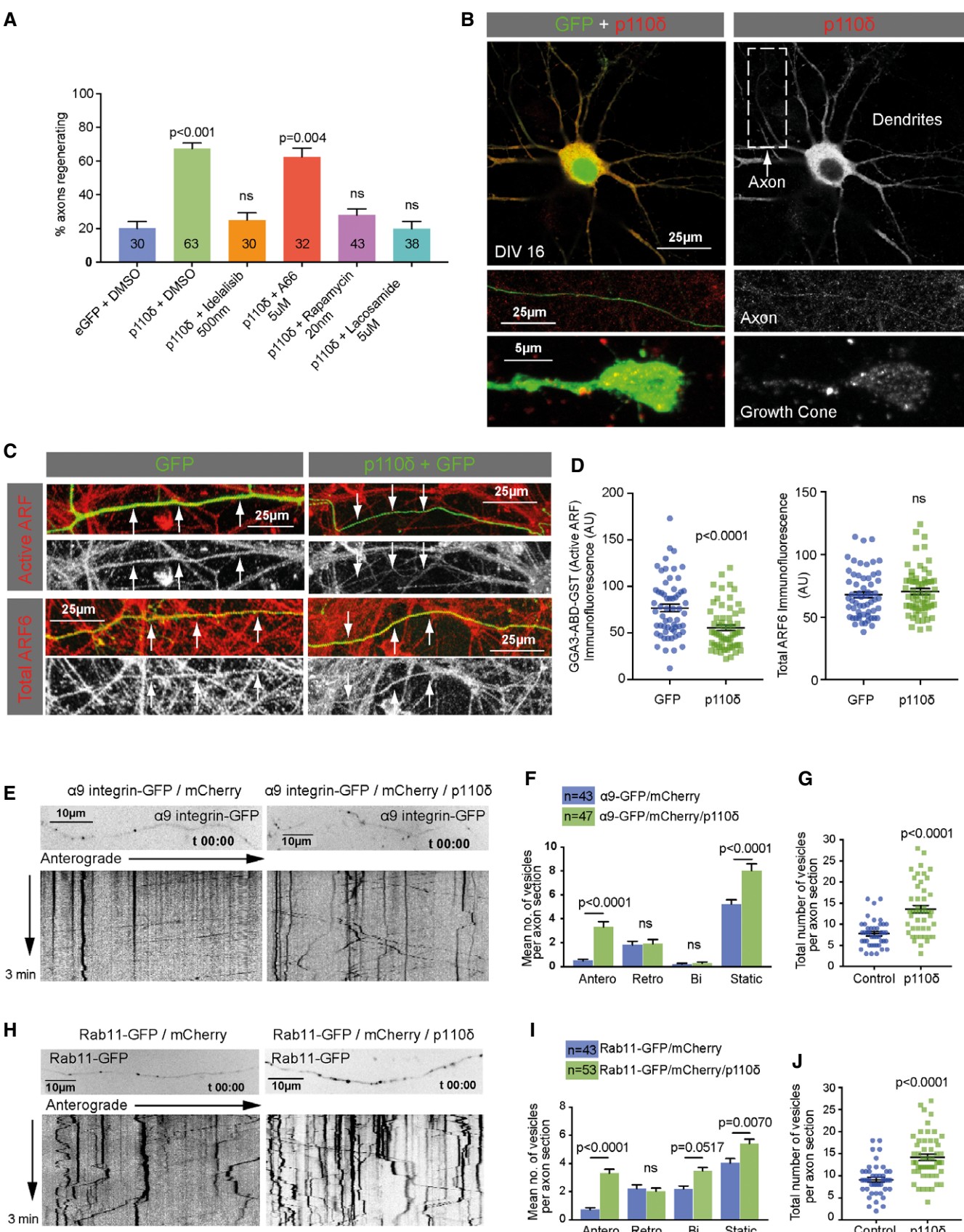

Figure 6.

**Figure 6. p110δ functions through multiple downstream pathways.**

A Percentage of regenerating axons after laser axotomy of DIV 14–17 rat cortical neurons expressing either GFP (control) or GFP and p110δ in the presence of the indicated inhibitors, 14h after laser injury. Numbers on bars are injured axons per group.

B Single confocal section through a DIV 16 neuron expressing p110δ and GFP (upper panels). Green colour is GFP, and red is overexpressed p110δ, detected with anti-p110δ. Lower panels show p110δ and GFP in a distal axon section, and at the growth cone. Arrow indicates the axon.

C Active ARF and total ARF6 (red) in axons of DIV 16 rat cortical neurons expressing GFP or GFP and p110δ, as indicated. Arrows indicate axons.

D Quantification of mean axonal ARF activity and total (mean) ARF6 in DIV 16 neurons expressing GFP or GFP and p110δ n = 60 axons for each condition.

E Kymographs showing dynamics of α9 integrin–GFP in the distal axons of DIV 14–16 neurons, control or co-transfected with p110δ.

F Quantification of α9 integrin–GFP dynamics in the distal axons of DIV 14–16 neurons, control or co-transfected with p110δ. n numbers are axon sections analysed for each condition.

G Quantification of total α9 integrin–GFP number in the distal axons of DIV 14–16 neurons, control or co-transfected with p110δ. n = 43 for control, 47 for co-transfected with p110δ. n numbers are axon sections analysed for each condition.

H Kymographs showing dynamics of Rab11–GFP in the distal axons of DIV 14–16 neurons, control or co-transfected with p110δ.

I Quantification of Rab11–GFP dynamics in the distal axons of DIV 14–16 neurons, control or co-transfected with p110δ. n numbers are axon sections analysed for each condition.

J Quantification of total Rab11–GFP in the distal axons of DIV 14–16 neurons, control or co-transfected with p110δ. n = 43 for control, 53 for co-transfected with p110δ. n numbers are axon sections analysed for each condition.

Data information: Data are shown as the mean ± SEM. (A) *P*-values indicate statistical significance analysed by Fisher's exact test. (F and I) *P*-values indicate significance measured by ANOVA with Tukey's *post-hoc* analysis. (D, G and J). Data were analysed by two-tailed Student's *t*-test.

expressing p110δ or by silencing PTEN, but that expression of p110δ has the most robust effects on axon regeneration. Together, these data show that enhancing PI3K activity in CNS neurons greatly enhances their ability to regenerate their axons and indicate viral delivery of p110δ to CNS neurons as a novel approach to boost signalling through this regenerative pathway.

# Discussion

Our study aimed to find a new method of stimulating axon regeneration based on manipulation of $PIP_3$ levels in neurons. In previous work, the approach to this has mainly been to knock down the $PIP_3$ dephosphorylating enzyme PTEN (Park *et al*, 2008; Liu *et al*, 2010; Lee *et al*, 2014; Geoffroy *et al*, 2015). However, this approach depends on generation of $PIP_3$ by PI3K, and if there is little $PIP_3$ being produced, PTEN knockdown will have little effect. We reasoned that limited $PIP_3$ production in mature neurons would explain the observation that PTEN knockout is effective at promoting axon regeneration in immature neurons, less so in the fully adult CNS. We used a new method to visualise $PIP_3$ levels in PNS and CNS neurons at different levels of maturity. In sensory axons, which regenerate readily, we observed high levels of $PIP_3$ in the growth cones, dynamically changing with growth cone movements. In cortical CNS neurons, developing immature neurons exhibited intense $PIP_3$ levels in their axon and growth cone during the period of rapid axon growth. However, $PIP_3$ levels decreased to a much lower level at a time when transport declines and axons lose their capacity for regeneration. We also investigated the role of PI3K isoforms in regeneration in PNS neurons using isoform-specific inhibitors. In sensory axons, which regenerate successfully and quickly, growing in microfluidic chambers to separate axonal and cell body compartments, inhibitors of p110α blocked both growth and post-axotomy regeneration when applied to either compartment, but a p110δ inhibitor blocked regeneration without affecting normal growth. This supports a previous study that links p110δ to sensory regeneration (Eickholt *et al*, 2007) and suggests that p110δ is uniquely linked to regeneration. Overexpression of p110 isoforms was therefore tested for the ability to increase $PIP_3$ levels and to promote regeneration.

Overexpression of p110δ in mature CNS neurons partially restored $PIP_3$ levels, particularly in axon tips but p110α was ineffective. However, introducing the hyperactivating H1047R mutation to p110α had the same effect on $PIP_3$ levels as p110δ.

Stimulation of axon regeneration was therefore assessed. p110δ and p110α$^{H1047R}$ transfected into mature cortical neurons strongly enhanced axon regeneration. In the CNS *in vivo*, transgenic expression led to enhanced neuroprotection in the retina, and regeneration in the optic nerve after a crush injury. Importantly, viral transduction of p110δ (which produces a higher level of expression) into adult RGC neurons led to axons regenerating for a greater distance after injury, demonstrating a novel approach to boost CNS regeneration through the PI3K pathway by gene transfer. Previous work has indicated that the hyperactive H1047R mutation of p110α can behave like p110δ to sustain AKT signalling in fibroblasts, by functioning independently of co-activation by Ras (Kang *et al*, 2006). We found that expression of native p110α had no effect on $PIP_3$ generation; it was only p110δ or p110α$^{H1047R}$ that enhanced axonal $PIP_3$ (Fig 3). This suggests that receptor activation of PI3K is at a low level in mature CNS neurons and that p110δ can potentiate PI3K signalling in low trophic conditions that are insufficient to activate native p110α. Our findings suggest that p110δ has a lower threshold of activation and that signals normally required to fully activate p110α (such as adhesion and growth factor receptors or Ras) may not be available in the axon. This has important implications for understanding the nature of p110δ beyond the nervous system and may help to explain how T cells utilise p110δ for development, differentiation and function (Okkenhaug, 2013).

The PI3K signalling pathway is a well-known regulator of axon regeneration, based on seminal studies which demonstrated that deletion of PTEN leads to robust regeneration in the CNS through downstream signalling via mTOR (Park *et al*, 2008). Aside from mTOR, the mechanism through which PTEN deletion stimulates regeneration is not completely understood. One difficulty in understanding its mechanism is that in addition to functioning as a lipid phosphatase (PTEN opposes PI3K by converting $PIP_3$ back to $PIP_2$), it also has protein phosphatase activity (Kreis *et al*, 2014). It is now also apparent that PTEN not only dephosphorylates $PI(3,4,5)P_3$, but also functions as a $PI(3,4)P_2$ phosphatase, a role linked with cancer

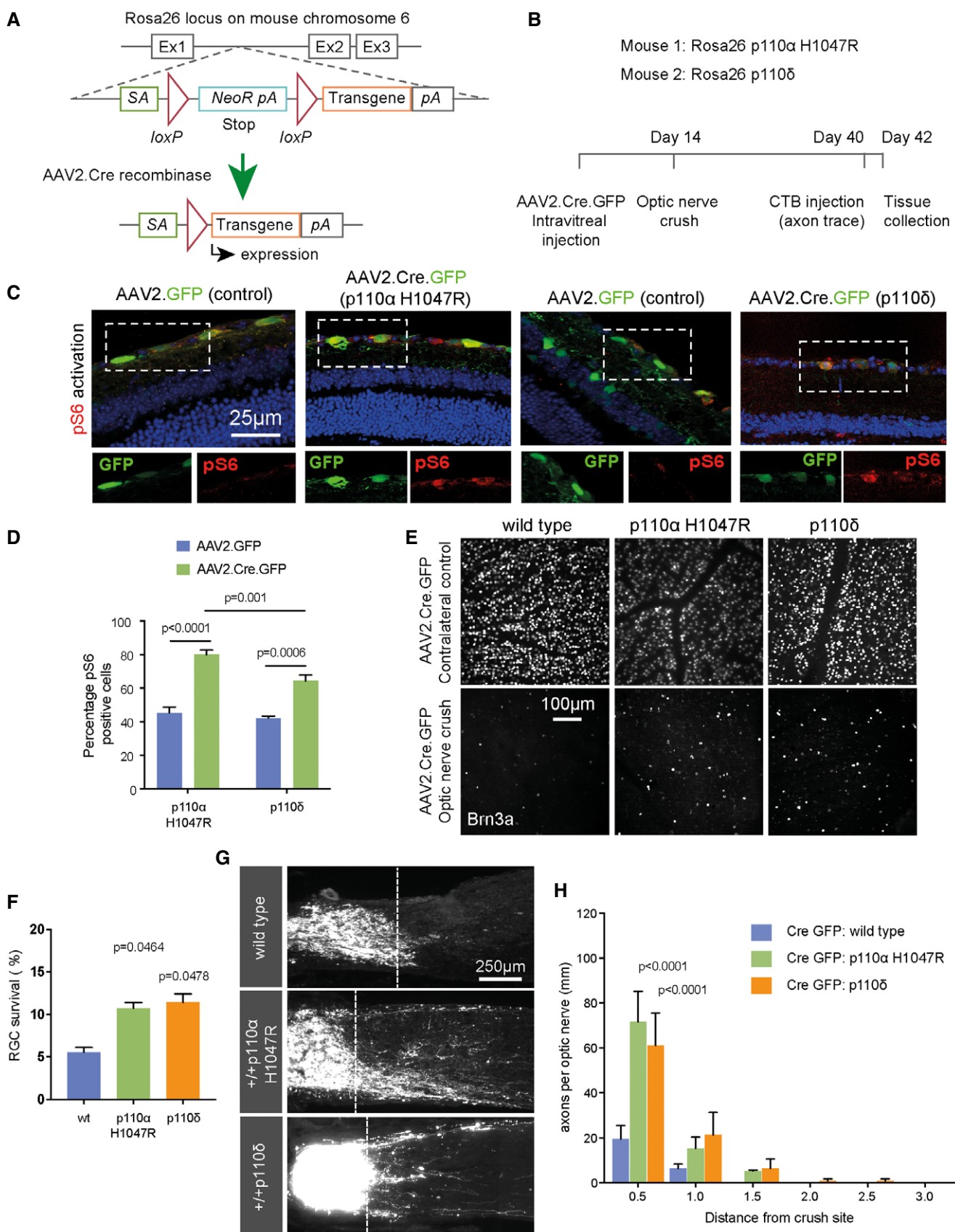

Figure 7.

**Figure 7.  p110δ and p110α$^{H1047R}$ support RGC survival and axon regeneration in the mouse optic nerve.**

A   Schematic representation of Rosa26 transgene expression via AAV.Cre recombinase.

B   Time course of optic nerve regeneration experiments using Rosa26 p110α$^{H1047R}$ and Rosa26 p110δ mice.

C   Retinal sections from AAV-injected mice, immunolabelled for phospho-S6. Blue colour is DAPI to indicate nuclei. Inset images show individual colours at the same scale as the full image.

D   Percentage of phospho-S6-positive cells in the RGC layer 2 weeks after delivery of AAV-Cre-GFP or AAV-GFP.

E   Retinal whole-mounts of AAV-injected mice immunolabelled for the RGC marker Brn3A to indicate cell survival.

F   Analysis of RGC survival 28 days after optic nerve crush (counts of Brn3A-positive RGC neurons).

G   Representative images of CTB-labelled RGC axons 4 weeks after optic nerve crush in wild-type (c57bl/6), Rosa26-p110α$^{H1047R}$ and Rosa26-p110δ transgenic mice injected with AAV2.Cre.GFP. Dashed line indicates crush site.

H   Regenerating axons at different distances distal to lesion sites.

Data information: (D, F and H) Data are shown as the mean ± SEM. *P*-values indicate significance as measured by ANOVA with Tukey's *post-hoc* analysis. n = 7 mice (wt), 10 mice (p110α$^{H1047R}$) and 6 mice (p110δ).

invasion (Malek *et al*, 2017). Our findings argue in favour of PTEN functioning through the regulation of PIP$_3$ and confirm the importance of this molecule in the regulation of axon regeneration.

Most PI3K signalling events rely on more than one isoform (Hawkins & Stephens, 2015), and whilst the p110δ isoform contributes to sciatic nerve regeneration (Eickholt *et al*, 2007), the contribution of the other isoforms remained unknown. Our findings demonstrate that both p110α and δ are required for efficient axon regeneration and that p110α functions in both the axon and cell body, whilst p110δ is specifically required in the axon (Fig 1). Inhibition of p110α opposed both axon growth and regeneration, whilst the action of p110δ was specific to regeneration, with the p110δ inhibitor blocking regeneration but not growth of uncut axons. This is in keeping with previous studies of the kinase-inactive p110δ$^{D910A}$ transgenic mouse, which has normal nervous system development, but defective PNS regeneration (Eickholt *et al*, 2007). Taken together, these results suggest that p110α mediates the somatic and axonal signalling that is necessary to support adult DRG axon growth (itself a regenerative phenomenon), whilst p110δ is further required within the axon to facilitate the redevelopment of a growth cone after injury. In CNS neurons, overexpression of either p110δ or p110α$^{H1047R}$ was sufficient to enable efficient regeneration both *in vitro* and *in vivo*, and AAV-mediated p110δ expression in RGCs promoted robust optic nerve regeneration.

Our studies into the downstream pathways mediating the effects of p110δ overexpression suggest the involvement of multiple pathways, including signalling through mTOR and CRMP2, which have previously been implicated in mediating the regenerative effects of PTEN deletion and PI3K activity (Park *et al*, 2008; Leibinger *et al*, 2019). Both of these pathways function downstream of AKT, which is recruited to PIP$_3$ by virtue of its PH domain. It is likely that additional pathways could contribute to regeneration downstream of either PIP$_3$ generation or PIP$_2$ reduction, due to the wide variety of proteins with PH domains, including the regulatory molecules of small GTPases that regulate the cytoskeleton, such as Rac1 and Cdc42 (Welch *et al*, 2002; Yoshizawa *et al*, 2005; Sosa *et al*, 2006), or that regulate axon transport such as ARF6 (Gillingham & Munro, 2007; Nieuwenhuis & Eva, 2018). Our findings suggest that p110δ exerts some of its effects by signalling through ARF6, a known regulator of integrin and Rab11 transport (Montagnac *et al*, 2009; Ghosh *et al*, 2019). These molecules contribute to the developmental loss of CNS regenerative ability, becoming transported away from axons and restricted to the dendrites and cell body as CNS neurons mature (Franssen *et al*, 2015; Koseki *et al*, 2017).

Overexpression of p110δ led to a restoration of anterograde transport of both integrins and Rab11 endosomes. The direction of Rab11 transport is likely regulated in a complex with ARF6, although it may potentially be regulated by activation state or by phosphoinositide generation (Campa & Hirsch, 2017). Previous work has shown that forcing Rab11 vesicles into these mature axons promotes regeneration (Koseki *et al*, 2017). Rab11 is also associated with the transport of other PI3K activating receptors, which are excluded from mature axons (Hollis *et al*, 2009a,b). Increased receptor transport due to p110δ expression suggests a feed-forward mechanism, in keeping with previous findings that BDNF stimulation leads to increased axonal transport of TrkB receptors during development (Cheng *et al*, 2011). Newly transported integrins could potentially signal through p110δ, as has previously been shown in PNS neurons (Eickholt *et al*, 2007) and cancer cells (Fiorcari *et al*, 2013).

The exclusion of PI3K-activating receptors from mature axons is probably a reason why the generation of PIP$_3$ is low in mature axons, and why overexpression of p110α in neurons did not increase PIP$_3$ levels. PTEN deletion and p110δ expression also enhance regeneration via mTOR signalling in the cell body; however, mTOR has recently been found to be present at the growth cone of developing axons in the mouse cerebral cortex, suggesting the PI3K-AKT-mTOR pathway may also function locally within the axon (Poulopoulos *et al*, 2019). Our findings demonstrate the importance of PIP$_3$ in the axon as well as the cell body for optimal regeneration. The importance of targeting the axon as well as the cell body is often overlooked; however, there is increased axonal transport as part of the PNS injury response, and in the CNS, increased transport enables regeneration (Petrova & Eva, 2018).

The search for methods for promoting regeneration in the damaged CNS continues. A successful strategy will likely involve multiple interventions. Could manipulation of PI3K be useful? PTEN deletion or p110α$^{H1047R}$ expression can be oncogenic in dividing cells. However, expression under a neuron-specific promoter targets expression to a non-dividing cell type, and it is unlikely that expression of p110δ in CNS neurons would lead to cell transformation, particularly given its usual expression in PNS neurons. Our study puts forward AAV-mediated delivery of p110δ as a novel means of stimulating regeneration through the PI3K pathway. We propose that p110δ should also be considered as an additional intervention with other regenerative strategies that target the PI3K pathway, either through growth factor treatments such as IGF-1 plus osteopontin (Duan *et al*, 2015; Liu *et al*, 2017), expression of activated

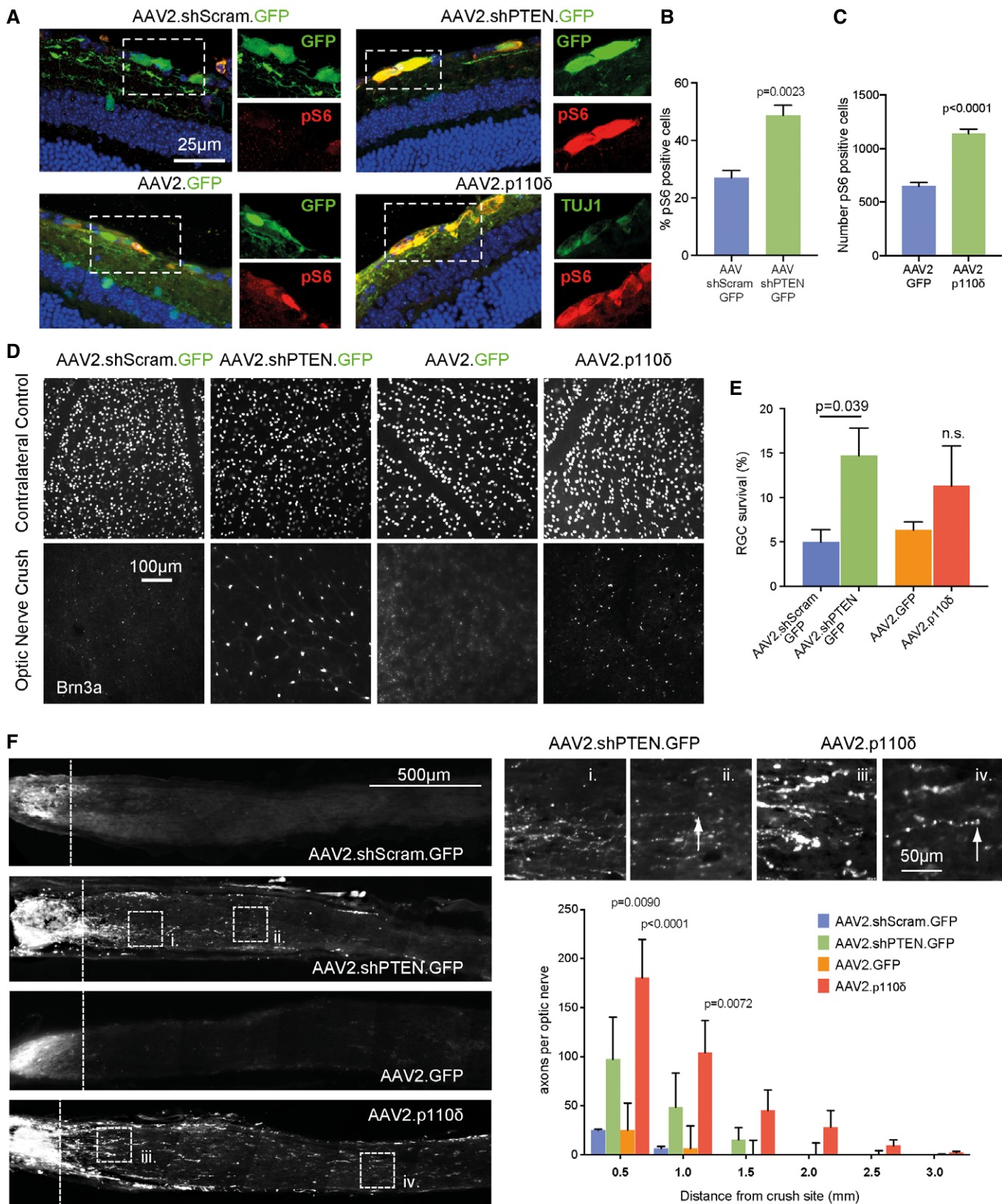

Figure 8.

**Figure 8.  AAV2-p110δ facilitates axon regeneration in the mouse optic nerve.**

A  Retinal sections from mice injected with AAV2 viruses as indicated, immunolabelled for phospho-S6. Blue colour is DAPI to indicate nuclei. Inset images show individual colours at the same scale as the full image.

B  Percentage of phospho-S6-positive cells in the retinal ganglion layer of mice injected with AAV2.shScramble.GFP or AAV2.shPTEN.GFP.

C  Number of pS6-positive RGCs of mice injected with AAV2.GFP or AAV2.p110δ.

D  Representative images of retinal whole-mounts from AAV-injected mice stained for the RGC marker Brn3A to indicate RGC survival.

E  RGC survival 28 days after optic nerve crush.

F  CTB-labelled RGC axons 4 weeks after optic nerve crush. Boxes i. to iv. show regenerating axons. Dashed line indicates crush site. White arrows mark the end of axons. Graph shows quantification of regenerating axons at different distances distal to the lesion sites.

Data information: (B) and (C) Data are shown as the mean ± SEM and P-values indicate significance measured by two-tailed Student's t-test. (E and H) Data are shown as the mean ± SEM and P-values indicate significance measured by ANOVA with Tukey's *post-hoc* analysis. n = 8 mice (scrambled shRNA control) 6 mice (shPTEN) 4 mice (GFP) and 7 mice (p110δ).

integrins (Cheah *et al*, 2016), or through pharmacological interventions such as insulin (Agostinone *et al*, 2018).

## Materials and Methods

### Mouse strains

C57BL6/J mice were used during this study, as well as four transgenic mouse strains: GFP-AKT-PH, Rosa26 p110α$^{H1047R}$, Rosa26 p110δ and B6;129S6-Gt(ROSA)26Sortm14(CAG-tdTomato)Hze/J (https://www.jax.org/strain/007908). The generation of the PIP3 reporter mouse GFP-AKT-PH has been previously described (Nishio *et al*, 2007) and was a gift from Dr Len Stephens (Babraham Institute, UK). Male and female

a gift from Jody Haigh (University of Manitoba, Canada; Nyabi *et al*, 2009). Targeting constructs harbouring sequences for p110α$^{H1047R}$ or p110δ were used to electroporate Bruce4 C57BL/6 ES cells. Selection of targeted clones was undertaken with G418, making use of the neomycin resistance selection cassette. Positively targeted clones were identified by Southern blotting. Positively targeted clones were microinjected into C57BL/6J-Tyrc-2J blastocysts by Babraham Institute Gene Targeting Facility (Babraham Institute, UK). These blastocysts were transferred to the oviducts of time-mated pseudopregnant foster mothers. The progeny were assessed for their degree of chimerism by coat colour and chimeric males mated to white C57BL/6J-Tyrc-2J females. Black progeny were tested for their correct incorporation of the Human PI3K genes at the Rosa26 locus by PCR using the following primers:

| Rosa26 | ROSA26 F1 AMQ | GGCTCAGTTGGGCTGTTTTG | WT allele = 359 bp |
|---|---|---|---|
| Hs p110 | ROSA26 R2 AMQ | TCTGTGGGAAGTCTTGTCCC | KI allele = 603 bp |
|  | ROSA26 loxP AMQ | GTGGATGTGGAATGTGTGCG |  |
| Hs p110α | Hs PIK3CA #5 intra fwd | TTGATCTTCGAATGTTACCT | KI allele = 813 bp |
|  | S2rev | CTTGATATCGAATTCCGCCCC |  |
| Hs p110δ | Hs PIK3CD #5 intra fwd | GTACTCCGTTCAGACACCAT | KI allele = 742 bp |
|  | S2rev | CTTGATATCGAATTCCGCCCC |  |

mice were used dependent on litters available with equal distributions across experiments. Rosa26 p110α$^{H1047R}$ and Rosa26 p110δ were generated according to a previously described protocol (Nyabi *et al*, 2009). Briefly, cDNA sequences for human p110α or p110δ were cloned into the pENTR D-TOPO vector (Thermo Fisher) to allow cloning into Gateway-compatible vectors. The H1047R point mutation was introduced by site-directed mutagenesis into PIK3CA using the QuickChange II XL kit (Stratagene #200521) and the primers QuickChange H1047R forward 5′-catgaaacaaatgaatgatgcacgccatggtggctggacaac-3′ and QuickChange H1047R reverse 5′-gttgtccagccaccatggcgtgcatcattcatttgtttcatg. The presence of the mutation was confirmed by sequencing prior to cloning into the destination vector. PIK3CD (p110δ) was a gift from Roger Williams (MRC Laboratory for Molecular Biology, UK). PIK3CA (p110α) was a gift from Bert Vogelstein (Addgene plasmid #16643; http://n2t.net/addgene:16643; RRID:Addgene_16643; Samuels *et al*, 2004). The Gateway®-compatible pROSA26 destination vector (pROSA26-DV1) was

### DRG culture

Dissociated DRG neuronal cultures were obtained from adult male Sprague Dawley (SD) rats and from the transgenic AKT-PH-GFP adult mouse. DRGs were incubated with 0.1% collagenase in Dulbecco's modified Eagle's medium (DMEM) for 90 min at 37°C followed by 10 min in trypsin at 37°C and dissociated by trituration. Dissociated cells were centrifuged through a layer of 15% bovine serum albumin (BSA) and cultured on 1 μg/ml laminin on glass-bottom dishes (Greiner) in DMEM supplemented with 10% foetal calf serum and 50 ng/ml nerve growth factor (NGF). For compartmentalised experiments, DRG neurons were plated in Xona microfluidic devices (Xona SND150) on glass coverslips. For PI3K inhibitor experiments, media were exchanged for serum-free media overnight. Separation of media was achieved by maintaining a pressure gradient between the axonal and somatic sides of the device.

### Embryonic cortical neuron culture

Primary rat cortical neuron culture has been described previously (Eva *et al*, 2017). Cultures were prepared from embryonic day 18 (E18) SD rats. Neurons were dissociated with papain for 8 min at 37°C, washed with HBSS and cultured in MACS Neuro Medium supplemented with MACS NeuroBrew (Miltenyi), and plated on glass-bottom dishes (Greiner) coated with poly-D-lysine. Cells were transfected at 10 DIV, and experiments were performed between DIV 14 and 17. Cortical neurons were transfected with magnetic nano-particles (Franssen *et al*, 2015).

### hESC dopaminergic neuron culture

RC17 hESC cell culture has been previously described in detail before (De Sousa *et al*, 2016; Koseki *et al*, 2017). Cells (RRID: CVCL_L206) were sourced from Roslin Cells, Scottish Centre for Regenerative Medicine, Edinburgh, UK. The cell line is free from mycoplasma contamination as determined by RT–qPCR. On d0, hESC were detached and transferred to form embryoid bodies from d0 to d4 in neural induction medium (Neurobasal:DMEM/F12 (1:1), 0.2% P/S, L-glutamine, N2, B27, recombinant Sonic Hedgehog Shh (C24II), recombinant noggin (Ng), SB431542 (SB) and CHIR99021 (CH)). Rock inhibitor (RI,) was present in the medium from d0 to d2. At d4, embryoid bodies were plated in poly-L-ornithine laminin and fibronectin (PLF)-coated plates and cultured in neural proliferation medium (Neurobasal:DMEM/F12 (1:1), 0.5xN2, 0.5xB27, supplemented with Shh, Ng, SB and CH from d4 to d7, and with Shh, Ng and CH from d7 to d9). At d11, cells were dissociated with Accutase and 50*104 cells per well single-cell suspensions were plated in PLF-coated 8-well chamber slides. Cells were cultured in neuronal differentiation medium (NB with 0.2% P/S, l-glutamine, B27, ascorbic acid, recombinant human BDNF, GDNF, db-CAMP; 50 mM) and DAPT from d14 onwards. Medium was replaced twice weekly up to day 50, after which medium was replaced weekly.

### N1E cell culture

N1E-115 cells (ATCC) were plated at a density of 5,000 cells/cm² in DMEM-high glucose with GlutaMAX (Gibco) + 10% FBS and incubated for 6–7 h at 5% $CO_2$ and 37°C.

### DNA constructs

Constructs for expressing p110 were generated from pHRsinUbEm, a bicistronic vector expressing EGFP under the control of the SFFV promoter and emerald fluorescent protein under the control of the Ubiquitin promoter. p110δ (PIK3CD) was a gift from Roger Williams (MRC Laboratory for Molecular Biology, UK). PIK3CD was cloned from pcDNA3.1 in place of EGFP. p110α (PIK3CA) or p110α H1047R was cloned into the same site from pBabe puro vectors. pBabe puro HA PIK3CA and p110α H1047R were gifts from Jean Zhao (Addgene plasmid #12522 and #12524; http://n2t.net/addgene:12522; RRID:Addgene_12522, http://n2t.net/addgene:12524; RRID:Addgene_12524) (Zhao *et al*, 2005). AAV.CAG.p110δ was generated by cloning the human PIK3CD sequence into an AAV-sCAG vector backbone by Gibson assembly (Gibson *et al*, 2009).

### Antibodies

The antibodies were as follows: PI(3,4,5)P: anti-PtdIns(3,4,5)P$_3$ monoclonal antibody (Z-P345b, 1:200, Echelon Biosciences), GFP (rabbit, 1:500 Abcam ab290), phospho-S6 ribosomal protein (Ser235/236) (91B2) (rabbit, 1:200, Cell Signaling 4857S), TUJ1 (βIII Tubulin) (mouse, 1:400, Promega G7121), PTEN (D4.3) XP (rabbit, 1:100, Cell Signaling 9188S), p110δ (rabbit, 1:500, Abcam ab1678), Brn3a (C-20) (goat, 1:200, Santa Cruz sc-31984). ARF6, (rabbit, 1:100, Abcam ab77581), GST (1:400, Abcam ab19256), anti-goat Alexa Fluor 647 (1:1,000, A21447, Life Technologies), anti-rabbit IgG-conjugated Alexa Fluor 568 (A10042, 1:1,000, Thermo Fisher Scientific), anti-mouse IgG-conjugated Alexa Fluor 568 (A11004, 1:1,000, Thermo Fisher Scientific), anti-rabbit IgG-conjugated Alexa Fluor 488 (A-21206, 1:1,000, Thermo Fisher Scientific) and anti-mouse IgG-conjugated Alexa Fluor 488 (A-21202, 1:1,000, Thermo Fisher Scientific).

### Small-molecule inhibitors

The following small-molecule inhibitors were used to inhibit the various isoforms of p110. The indicated concentrations were chosen based on their known IC50 and from previously reported cell culture experiments: Pan-p110 (p110α/β/δ): LY294002, 20 μM (IC50 (in cell-free assays) 0.5 μM/0.97 μM/0.57 μM, respectively); p110α: A66, 5 μM (IC50 32 nM); p110α/δ: XL-147, 5 μM (IC50 39 nM/36 nM); p110β: TGX221, 500 nM (IC50 5 nM); p110δ: IC-87114, 10 μM (IC50 0.5 μM); p110δ: idelalisib, 500 nM (IC50 2.5 nM), lacosamide, 5 μm (Leibinger *et al*, 2019), and rapamycin, 20 nm (IC50 0.1 nM). DMSO was added at a volume corresponding to volume of inhibitor used in the experimental condition, to a maximum of 5 μl (in a final volume of 3 ml culture media).

### Virus production and injection

Three viruses were sourced commercially: AAV2.CMV.Cre.GFP (Vector Biolabs, Catalog #7016), AAV2.CMV.GFP (Vigene Biosciences, Catalog #CV10004) and AAV2.U6.shRNA (scramble).CMV.GFP (SignaGen Laboratories, Catalog #SL100815). AAV2.U6.shPTEN.CMV.GFP was a gift from Zhigang He (Boston Children's Hospital), and AAV2.CAG.p110delta was produced by Vigene Biosciences. Mice received 2 μl intravitreal injections of AAV. All viruses were injected into the left eye only at $1 \times 10^{13}$ GC/ml. For validation experiments, mice also received intravitreal injection of the appropriate control into the right eye.

### Optic nerve injury

Optic nerve injuries were carried out as previously described (Smith *et al*, 2009). The optic nerve behind the left eye was exposed intraorbitally and crushed with fine forceps for 10 s, approximately 0.5 mm behind the optic disc. Twenty-six days after the injury, mice received a 2 μl intravitreal injection of cholera toxin subunit β (CTB) with an Alexa Fluor 555 conjugate at 1 mg/ml. Twenty-eight days post-crush, animals were perfused with 4% paraformaldehyde (PFA) and the eyes and optic nerves collected for analysis. Surgical procedures were performed under anaesthesia using intraperitoneal injection of ketamine (100 mg/kg) and xylazine (10 mg/kg). This

research was regulated under the Animals (Scientific Procedures) Act 1986 Amendment Regulations 2012 following ARRIVE guidelines and ethical review by the University of Cambridge Animal Welfare and Ethical Review Body (AWERB). We also followed the Association for Research in Vision and Ophthalmology's Statement for the Use of Animals in Ophthalmic and Visual Research. Research was carried out on UK Home Office Project Licence 70/8152 under protocols 3 (breeding) and 4 (ocular injections and injury). Mice had unrestricted access to food and water and were maintained on a 12-h light/dark cycle in groups of five. Adult (2–6 months) male and female mice were used dependent on litters available with equal distributions across experiments. The sample size used in each experiment is stipulated in the figure legends.

## Immunohistochemistry

Retinas were fixed in 4% PFA for 2 h. Whole-mounts were washed four times with 0.5% PBS-Triton X-100. In between the second and third wash, a permeation step was performed to improve antibody penetration by freezing the retinas in 0.5% PBS-Triton X-100 for 10 min at −70°C, and washing was continued after thawing. Optic nerves were fixed overnight at 4°C in 4% PFA, followed by 30% sucrose overnight at 4°C. Sections were blocked with 2% BSA and 10% donkey serum in 2% PBS-Triton X-100. Primary antibodies were incubated at 4°C overnight and secondary antibodies for 2 h at room temperature.

## Standard immunocytochemistry

Cortical neurons were fixed with 3% paraformaldehyde (PFA) in PBS for 15 min and permeabilised with 0.1% Triton X-100 in PBS for 5 min. Cells were blocked with 3% bovine serum albumin (BSA) in PBS for 1 h. After blocking, the cells were incubated with primary antibodies diluted in 3% BSA in PBS at 4°C overnight and secondary antibodies that were diluted in 3% BSA in PBS for 1 h at room temperature.

## Phospholipid fixation and immunocytochemistry

We adapted a fixation technique previously used to detect $PI(4,)P_2$ and $PI(4,5)P_2$ (Hammond et al, 2009). Cortical neuron cultures were fixed using pre-warmed (37°C) 3% formaldehyde and 0.2% glutaraldehyde (GA; G011/3, TAAB Laboratories) in PBS for 15 min at room temperature, washed in 50 mM $NH_4Cl$ in PBS and then maintained at 4°C. Cells were incubated with 4°C blocking and permeabilisation solution (0.2% saponin, 50 mM $NH_4Cl$ and 3% BSA in PBS) for 30 min and then with anti-PtdIns(3,4,5)P3 (Z-P345b, 1:200, Echelon Biosciences) in blocking solution for 3 h before washing three times in 50 mM $NH_4Cl$ for 30 min. The secondary antibody was applied for 2 h at 4°C. After another 30-min wash, cells were post-fixed in 3% formaldehyde in PBS for 5 min at 4°C before being moved to room temperature for a further 10 min.

## Axonal ARF6 activation assay

Active ARF protein was detected using a previously established protocol (Eva et al, 2017) by means of a peptide derived from the active ARF-binding domain (ABD) of GGA3 fused to a GST tag

(GGA3–ABD–GST, Thermo Fisher Scientific). Neurons were fixed for 15 min in 3% formaldehyde (TAAB) in PBS, permeabilised with 0.1% Triton X-100 for 2 min and incubated with 20 µg/ml GGA3–ABD–GST in TBS and 1 mM EDTA overnight at 4°C. The GST tag was then detected using rabbit anti-GST antibody (ab19256, Abcam, 1:400). Control and experimental cultures were fixed and labelled in parallel, using identical conditions. Axons were analysed at 200–1,000 µm distal to the cell body. Images of axons were acquired by confocal laser-scanning microscopy using a Leica TCS SPE confocal microscope. Detection settings were adjusted so that the pixel intensities of acquired images were below saturation. Settings were then stored and were applied for the identical acquisition of each image using Leica LAS AF software. Lines were then traced along sections of axons to define the region of interest, and mean pixel intensities per axon section were quantified using Leica LAS AF. Images were corrected for background by subtracting an identical adjacent region of interest. The same technique was then used for measuring total levels of ARF6 in axons, after ARF6 immunolabelling.

## Insulin stimulation of N1E cells

Cells were starved in DMEM without serum overnight in the presence of either 500 nM GDC-0941 (Selleckchem) or an equal volume of cell culture grade DMSO (AppliChem). Cells were stimulated with 20 µg/ml insulin (Sigma) in DMEM or control treated with equal amounts of DMEM for 1 min prior to fixation in $PIP_3$-Immobilisation fixative and stained with for $PIP_3$ (anti-$PIP_3$, Echelon) and F-actin (Phalloidin, Thermo Fisher Scientific).

## Confocal and widefield microscopy

Laser-scanning confocal microscopy was performed using a Leica DMI4000B microscope and a Leica TCS SPE confocal system controlled with Leica LAS AF. Fluorescence and widefield microscopy were performed using a Leica DMI6000B with a Leica DFC350 FX CCD camera and a Leica AF7000 with a Hamamatsu EM CCD C9100 camera and Leica LAS AF. Leica AF7000 was used for imaging of axon and growth cone regeneration after axotomy.

## TIRF microscopy

Total internal reflection fluorescence (TIRF) microscopy was carried out on a Leica DMI6000B adapted with a dedicated TIRF module from Rapp OptoElectronic. GFP was excited with a 488 nm laser, and images were acquired using a Leica Plan Apo 100×/NA1.47 Oil TIRF objective and Hamamatsu EM CCD C9100 camera controlled by Leica LAS AF and Rapp OptoElectronic software.

## Laser axotomy of DRG neurons

Laser axotomy of DRG neurons was performed as described previously (Eva et al, 2017). Axons were cut directly before a growth cone, to determine the proportion of axons that regenerate rapidly after injury. Cultures were serum-starved overnight, and inhibitors were added at the start of the experiment. Preliminary experiments confirmed there was no effect of DMSO on the rate of regeneration, by comparing DMSO vs. no DMSO conditions. Axons were severed using a 355 nm DPSL laser (Rapp OptoElectronic, Hamburg,

Germany) connected to a Leica DMI6000B, and images were acquired every 15 min for 2 h. Successful regeneration was determined as the development and extension of a new growth cone.

### Laser axotomy of cortical neurons

Laser axotomy of cortical neurons was performed as described previously (Eva *et al*, 2017; Koseki *et al*, 2017) Axons were severed using a 355 nm DPSL laser (Rapp OptoElectronic, Hamburg, Germany) connected to a Leica DMI6000B. Cortical neurons were axotomised at DIV 14–17 at distances of 800–2,000 μm from the cell body on a section of axon free from branches. Images after axotomy were acquired every 30 min for 14 h. Regeneration was classed as the development of a new growth cone followed by axon extension for a minimum of 50 μm.

### Laser axotomy of hESC-derived neurons

Laser axotomy of hESC-derived neurons was performed as described previously (Koseki *et al*, 2017). Axons were severed using a 365 nm laser (MicroPoint, Andor) connected to an Andor spinning disk confocal microscope. Neurons were axotomised at d45–65 at distances of > 500 μm distal to the cell body on a section of axon free from branches. A single axon cut was made per neuron. Images after axotomy were acquired every 20 min for 16 h. Regeneration was classed as the development of a new growth cone followed by axon extension for a minimum of 50 μm.

### Quantification of PIP$_3$ in cortical neurons

E18 cortical neurons were fixed at DIV 3, 8 or 16, and PIP$_3$ was detected as described above. All cultures were fixed and labelled using identical conditions. Images were acquired by confocal laser-scanning microscopy using a Leica TCS SPE confocal microscope. Identical settings were used for each image using Leica LAS AF. Z-stacks were acquired for each image, spanning the entire depth of either the cell body or the growth cone. PIP$_3$ fluorescence intensity was measured using Leica LAS AF. Immunofluorescence intensities were calculated by measuring the region of interest (ROI) and subtracting the intensity of a control region adjacent to the ROI.

### Quantification of PIP$_3$ in N1E cells

Imaging was performed on a Leica SP5 confocal system with a 40× objective with identical settings across experiments. Stacks covered the complete height of all cells with. Areas with comparable cell densities were selected in the F-actin channel. Mean PIP$_3$ intensity in N1E cells was quantified in Fiji (ImageJ 1.51n) on max projections. Intensities were compared in 15 images per treatment from three independent cultures.

### Analysis of neuronal morphology

Images were captured on a Leica DMI6000B, with a 40X-oil objective using Leica LAS AF. Semi-automated and standardised analysis was performed using MATLAB platform version 2017 and SynD (Schmitz *et al*, 2011). The output of SynD was used for data analysis of dendritic length, fluorescent intensities, sholl analysis and soma size.

### Neurite outgrowth assay

Cultured cortical neurons were transfected at 2 DIV and fixed at 4 DIV. Images for neuronal morphology were acquired using a Leica DMI6000B microscope, with a 40X-oil objective using Leica LAS AF. Neurite lengths were measured using the ImageJ plugin "simple neurite tracer" (Longair *et al*, 2011).

### Axon transport analysis

Axon transport analysis of integrins and Rab11 has been described in detail before (Franssen *et al*, 2015; Eva *et al*, 2017). Briefly, cortical neurons were transfected at DIV 10 with α9 integrin–GFP, or Rab11–GFP together with either mCherry (control) or mCherry plus p110δ, and imaged at DIV 14–16 DIV using spinning disc confocal microscopy. Sections of axons were imaged at a region in the distal part of the axon (> 800 μm from the cell body). Vesicles were tracked for their visible lifetime and analysed by kymography to classify the proportion of vesicles classed as anterograde, retrograde, bidirectional or immobile per axon section. Vesicles with a total movement less than 2 μm during their visible lifetimes were classed as immobile. Vesicles moving in both directions but with net movement of < 2 μm (during their visible lifetimes) were classed as bidirectional. Vesicles with net movements > 5 μm in either direction by the end of their visible lifetimes were classed as anterograde or retrograde accordingly.

### Optic nerve regenerating axon counts

To measure regenerating RGC axons after optic nerve crush, longitudinal sections of optic nerves were serially collected. Regenerating RGC axons were quantified as described previously (Smith *et al*, 2009), by counting the number of CTB-labelled axons at the indicated distances beyond the crush site from four sections per optic nerve. Axonal sections were imaged using a Zeiss AxioScan Z1 at ×40 magnification.

### RGC survival counts

For retinal whole-mounts, two images were taken from each of the four retinal quadrants at ×20 magnification, sampling both the more central and the more peripheral region of each quadrant. Images were then analysed in ImageJ Fiji using Image-Based Tool for Counting Nuclei (ITCN) Plugin (University of California, Santa Barbara, CA, USA) to count Brn3A-labelled cells. The number of RGCs in the left injured eye was expressed as a percentage survival value compared to the mean number of RGCs of the contralateral control eyes.

### RGC fluorescence analysis

To confirm successful viral transduction, eye cup images were stained for GFP or PI3KDelta counterstained with DAPI and imaged using a Zeiss AxioScan Z1 at ×20 magnification. Retinal sections immunolabelled for pS6 were examined by fluorescence

**The paper explained**

**Problem**

Young neurons in the central nervous system (CNS) can regrow their axons after injury, but this ability is lost as they mature. Axonal injury or disease in the adult brain, eyes and spinal cord therefore has devastating consequences, and can result in neurological impairment, vision loss or paralysis. Conversely, neurons of the peripheral nervous system (PNS) maintain the ability to regenerate their axons into adulthood. Comparing PNS and CNS neurons is one approach to identifying new ways of promoting injured CNS axons to regenerate after injury.

**Results**

Our study found that adult PNS neurons use two versions of the enzyme PI 3-kinase to regenerate their axons, p110δ and p110α. These enzymes generate the phospholipid PIP$_3$. Visualisation of PIP$_3$ in maturing CNS neurons revealed that PIP$_3$ is strongly downregulated at the time when these neurons lose their regenerative ability. Overexpression of p110δ elevated axonal PIP$_3$ and regeneration after laser injury, but p110α required the activating H1047R mutation to behave similarly, demonstrating that native p110δ functions in a hyperactive fashion. The study found that p110δ functioned through multiple mechanisms, including the enhanced transport of regenerative machinery into axons. Importantly, viral delivery of p110δ to the retina promoted axon regeneration after an optic nerve crush injury and was accompanied by enhanced survival of RGC neurons in the retina.

**Impact**

These findings suggest that signalling through PI 3-kinase-linked receptors is limited in adult CNS axons, contributing to their weak regenerative ability. Exogenous expression of the p110δ subunit elevates axonal PI 3-kinase activity by functioning in a hyperactive fashion, leading to enhanced regeneration in human and rodent models of CNS injury, and enhanced neuroprotection and regeneration after a mouse optic nerve crush injury.

microscopy. One hundred GFP-positive RGCs were counted and identified as positive or negative for pS6. For AAV2.p110δ-transduced RGCs, the total number of pS6-positive cells was counted from 12 retinal sections throughout the eye compared to total number of pS6-positive cells from the control group using TUJ1 to identify neurons. Fluorescence intensity was measured using ImageJ Fiji.

**Quantitative and statistical analysis**

Statistical analysis was performed throughout using GraphPad Prism. Fisher's exact test was calculated using GraphPad online (https://www.graphpad.com/quickcalcs/contingency1.cfm). Percentage of regenerating axons was compared by Fisher's exact test. Sample size calculations using data from previous experiments and pilot studies were used to minimise the group sizes required to demonstrate clinically relevant (as opposed to simply statistically significant) effect sizes. Animals of both sexes were chosen randomly, different viral treatments were assigned to animals randomly, and analysis and quantification were blinded. Normality was tested for with GraphPad Prism software, using the D'Agostino–Pearson normality test, when appropriate. Kruskal–Wallis and Fisher's exact test were used as indicated in the

manuscript. Variation was tested using the Brown–Forsythe test using GraphPad Prism. Comparisons between more than two experimental groups (e.g. expression of various PI3K isoforms) and one measured variable were done using one-way ANOVA to test for variance and Tukey's multiple comparison test.

# Data and software availability

This study includes no data deposited in external repositories.

**Expanded View** for this article is available online.

## Acknowledgements

We acknowledge the Babraham Gene Targeting Facilities for the generation of the Rosa26 p110 mouse strains and would like to thank Dr. Len Stephens and Dr. Phill Hawkins (Babraham) for the gift of the GFP-AKT-PH mouse strain, and for their advice and support. RE, JWF, KRM, ACB, BN, RSE and CSP were supported by grants from MRC-Sackler, International Spinal Research Trust (NRB110), Wellcome Trust, ERA-NET NEURON grant AxonRepair, the European Union, the Operational Programme Research, Development and Education in the framework of the project "Centre of Reconstructive Neuroscience", Czech Ministry of Education, CZ.02.1.01/0.0./0.0/15_003/0000419, Medical Research Council MRC (MR/R004544/1, MR/R004463/1), Christopher and Dana Reeve Foundation, Cambridge Eye Trust, Fight for Sight, and National Eye Research Council and a core support grant from the Wellcome Trust and MRC to the Wellcome Trust—Medical Research Council Cambridge Stem Cell Institute. BJE funding was provided by the DFG (SFB 958, TP16; SFB TRR 186, TP10). SvE was funded by EMBO ALTF 1436-2015, and SvE and Cff-C by MS Society UK. KO laboratory funding was provided by the BBSRC (BBS/E/B/000C0409, BBS/E/B/000C0427) and the Wellcome Trust (095691/Z/11/Z).

## Author contributions

BN, ACB, RSE, CSP, JF, PDS, SE and RE performed and analysed experiments. ARM generated PI3K transgenic mice. BH, AO and LAH generated AAV.p110-delta. TZK, RC, JC and SSD analysed RGC survival. KO designed and supervised transgenic generation of mice. BJE designed and supervised PIP$_3$ detection experiments. RE, JWF, KRM, ACB, TZK, CF-C, SE, BJE and KO conceived designed and supervised experiments and obtained funding. RE, BN, ACB and RSE prepared movies and figures. RE wrote the manuscript.

## Conflict of interest

KO received consultancy payments and/or research funding from Karus Therapeutics, Gilead Sciences and GlaxoSmithKline.

## For more information/relevant links

This study focuses on regeneration in the injured optic nerve, as part of projects aimed at repairing the damage that can occur in the optic nerve due to degenerative conditions such as glaucoma. It is also part of major research projects aimed at stimulating axon regeneration in the injured spinal cord. Below are links to author websites, charities and patient associations relevant to these conditions.

(i)    https://www.researchgate.net/profile/Bart_Nieuwenhuis3
(ii)   https://www.researchgate.net/profile/Amanda_Barber
(iii)  https://www.researchgate.net/profile/Rachel_Evans45
(iv)   https://www.researchgate.net/profile/Richard_Eva
(v)    International Glaucoma Association: www.glaucoma-association.com
(vi)   Centre for Eye Research Australia: www.cera.org.au

(vii)   Fight for Sight: https://www.fightforsight.org.uk/
(viii)  International Spinal Cord Society: https://www.iscos.org.uk/
(ix)    Christopher and Dana Reeve Foundation: https://www.christopherreeve.org/
(x)     Spinal Research: https://www.spinal-research.org/
(xi)    Wings for Life: https://www.wingsforlife.com/en/

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
