## [Review Process File · EMBO Molecular Medicine]

PI 3-kinase delta enhances axonal PIP3 to support axon regeneration in the adult CNS

Bart Nieuwenhuis, Amanda Barber, Rachel Evans, Craig Pearson, Joachim Fuchs, Amy MacQueen, Susan Erp, Barbara Haenzi, Lianne Hulshof, Andrew Osborne, Raquel Conceicao, Tasneem Khatib, Sarita Deshpande, Joshua Cave, Charles Ffrench-Constant, Patrice Smith, Klaus Okkenhaug, Britta Eickholt, Keith Martin, James Fawcett, and Richard Eva

DOI: 10.15252/emmm.201911674

Corresponding author(s): Richard Eva (re263@cam.ac.uk)

Review Timeline:

Submission Date:	4th Nov 19
Editorial Decision:	28th Nov 19
Revision Received:	27th Apr 20
Editorial Decision:	17th May 20
Revision Received:	27th May 20
Accepted:	29th May 20

Editor: Celine Carret

Transaction Report:

28th Nov 2019

Dear Dr. Eva,

Thank you for the submission of your manuscript to EMBO Molecular Medicine. We have now heard back from the three referees whom we asked to evaluate your manuscript.

You will see that all three find the study to be of putative interest and importance for the field of axonal regeneration. Still, both referees 2 and 3 want to see more mechanism. Referee 1 suggests using different cells to not exclusively rely on over expression means, while also suggesting a few more additional experiments that if performed would strengthen the findings. Finally, referee 3 requests a functional assay, which we agree is needed to claim any medical relevance.

We would therefore welcome the submission of a revised version within three months for further consideration and would like to encourage you to address all the criticisms raised as suggested to improve conclusiveness and clarity. Please note that EMBO Molecular Medicine strongly supports a single round of revision and that, as acceptance or rejection of the manuscript will depend on another round of review, your responses should be as complete as possible.

I look forward to receiving your revised manuscript.

Yours sincerely,

Celine Carret

Celine Carret, PhD
Senior Editor
EMBO Molecular Medicine

*** Instructions to submit your revised manuscript ***

**** PLEASE NOTE **** As part of the EMBO Publications transparent editorial process initiative (see our Editorial at <https://www.embopress.org/doi/pdf/10.1002/emmm.201000094>), EMBO Molecular Medicine will publish online a Review Process File to accompany accepted manuscripts.

To submit your manuscript, please follow this link:

Link Not Available

- 1) a .doc formatted version of the manuscript text (including Figure legends and tables). Please make sure that the changes are highlighted to be clearly visible to referees and editors alike.
- 2) separate figure files*
- 3) supplemental information as Expanded View and/or Appendix. Please carefully check the authors guidelines for formatting Expanded view and Appendix figures and tables at <https://www.embopress.org/page/journal/17574684/authorguide#expandedview>
- 4) a letter INCLUDING the reviewers' reports and your detailed responses to their comments (as Word file)

Also, and to save some time should your paper be accepted, please read below for additional information regarding some features of our research articles:

- 5) The paper explained: EMBO Molecular Medicine articles are accompanied by a summary of the articles to emphasize the major findings in the paper and their medical implications for the non-specialist reader. Please provide a draft summary of your article highlighting
 - the medical issue you are addressing,
 - the results obtained and
 - their clinical impact.

- 6) For more information: There is space at the end of each article to list relevant web links for further consultation by our readers. Could you identify some relevant ones and provide such information as well? Some examples are patient associations, relevant databases, OMIM/proteins/genes links, author's websites, etc...

7) Author contributions: the contribution of every author must be detailed in a separate section (before the acknowledgments).

8) EMBO Molecular Medicine now requires a complete author checklist (<https://www.embopress.org/page/journal/17574684/authorguide>) to be submitted with all revised manuscripts. Please use the checklist as a guideline for the sort of information we need WITHIN the manuscript as well as in the checklist. This is particularly important for animal reporting, antibody dilutions (missing) and exact p-values and n that should be indicated instead of a range.

9) Every published paper now includes a 'Synopsis' to further enhance discoverability. Synopses are displayed on the journal webpage and are freely accessible to all readers. They include a short stand first (maximum of 300 characters, including space) as well as 2-5 one sentence bullet points that summarise the paper. Please write the bullet points to summarise the key NEW findings. They should be designed to be complementary to the abstract - i.e. not repeat the same text. We encourage inclusion of key acronyms and quantitative information (maximum of 30 words / bullet point). Please use the passive voice. Please attach these in a separate file or send them by email, we will incorporate them accordingly.

You are also welcome to suggest a striking image or visual abstract to illustrate your article. If you do please provide a jpeg file 550 px-wide x 400-px high.

10) A Conflict of Interest statement should be provided in the main text

11) Please note that we now mandate that all corresponding authors list an ORCID digital identifier. This takes <90 seconds to complete. We encourage all authors to supply an ORCID identifier, which will be linked to their name for unambiguous name identification.

Currently, our records indicate that the ORCID for your account is 0000-0003-0305-0452.

Link Not Available

12) The system will prompt you to fill in your funding and payment information. This will allow Wiley to send you a quote for the article processing charge (APC) in case of acceptance. This quote takes into account any reduction or fee waivers that you may be eligible for. Authors do not need to pay any fees before their manuscript is accepted and transferred to our publisher.

Photos 400-800 DPI

Figures are not edited by the production team. All lettering should be the same size and style; figure panels should be indicated by capital letters (A, B, C etc). Gridlines are not allowed except for log plots. Figures should be numbered in the order of their appearance in the text with Arabic numerals.

Each Figure must have a separate legend and a caption is needed for each panel.

*Additional important information regarding figures and illustrations can be found at <http://bit.ly/EMBOPressFigurePreparationGuideline>

***** Reviewer's comments *****

Referee #1 (Remarks for Author):

This is an interesting manuscript looking at the potential role of PI3K-alpha and -delta isoforms in axonal regeneration. Their investigation pertains to the difference in regenerative capability of the peripheral nervous system compared to the central nervous system. They found that peripheral neurons, rely on both PI3Kalpha and delta to generate PtdIns(3,4,5)P3 for axonal regeneration whereas PtdIns(3,4,5)P3 levels are greatly reduced following development in central neurons. Interestingly they found that only expression of p110 δ restored axonal PtdIns(3,4,5)P3 level which increases regenerative axon transport and enhance CNS regeneration in multiple systems and viral expression of p110delta was sufficient to promote regeneration following optic nerve injury. Although the manuscript is generally sound, I see a number of control issues and one general conceptual problem that will need to be addressed either experimentally or at the very least discussed.

Major issues

1- The investigators should use the p110deltaD910A/D910A mouse line and extract neurons which would make this investigation a lot stronger as it would not rely on pharmacology and over-expression only. If the investigators do not have access to this line, at the very least, the authors should use over-expression of p110deltaD910A to assess the effect of blocking PI3K δ activity in all experiments relying on overexpression of PI3Kdelta. Similarly, the specific effects of p110alpha or delta over-expression should be tested using appropriate inhibitor cocktails as in Fig. 1. Finally, no additional effect of inhibitors should be detected upon overexpression of kinase-dead p110deltaD910A.

2- Although type-I PI3K isoforms are expected to act at the level of the plasma membrane, the actual distribution of the delta isoform is in the Golgi. This was shown in macrophages (Low et al., JCB 2010) in cortical neurons and microglia (Low et al., Nat Com 2014). In the Golgi, PI3K δ controls the anterograde trafficking of cytokines such as TNF α (in microglia) and of APP (in neurons) (Martínez-Mármol et al., J Neurosci., 2019). It is highly surprising that none of these papers are cited or discussed especially considering that some of the results of the manuscript (integrins and Rab11 anterograde transport) actually nicely corroborate these previous reports. Further, the fact that integrins is affected by PI3Kdelta inhibitor was also published previously and not cited in this manuscript (Fiorcari et al., PloS One 2013). The authors barely mentioned this as a possibility and instead use relatively weak correlative evidence to infer that increased axonal PtdIns(3,4,5)P3 can restore the anterograde transport of integrins as well as the Rab11 in endosomes. It is unclear to me how PI3Kdelta activity promotes these changes at the level of the Golgi and how? This should either be tested or at the very least discussed. Can over-expression of AKT-PH affect such transport? Can AKT-PH bind to the Golgi and colocalize with PI3Kdelta? This could potentially be tested directly on the trafficking of integrins and Rab11.

3- In the same vein, the fact that PtdIns(3,4,5)P3 correlate with the ability of neurons to regenerate is an interesting point of the paper but I feel it could be tested further using PTEN inhibitors. VO-OHpic has previously been used with success to show that PI3Kdelta and PTEN controls the ratio

between PtdIns(3,4,5)P3 and PtdIns(4,5)P2 in neurosecretory cells (Wen et al., Nature Com 2011). It should be used to see if it can increase regeneration in a PI3K α /delta-dependent manner.

Minor points:

- Please use the correct phosphoinositides nomenclature throughout the paper. PtdIns(3,4,5)P3 is spelled in 3 different ways in the paper.
- The concentration of inhibitors is correctly used and stated but it would be nice not to have to check in methods to get the info. Concentrations and timing should be included in the legend.
- Because, several concentrations of drugs have been used, it is not clear how much DMSO has been used. It should be stated and the authors should check that there is no effect of DMSO at such concentration (inclusion of a no DMSO condition).
- CAL101/Idalisib tends to fall out of solution in buffers. How did the author assess the correct concentration?
- The authors should harmonize letter font and size across all the figures of the manuscript.

Referee #2 (Remarks for Author):

In this paper the authors presented a very ambitious and interesting idea regarding the difference in axon regeneration capacity between CNS and PNS, from the viewpoint of PI3K biology. First the authors have postulated that PNS-specific expression of p110 δ and p110 α -CA support axonal regeneration and CNS neurons could gain this property by their overexpression. As a molecular mechanism, the authors showed axonal transport of α 9-integrin and Rab11 is enhanced by p110 δ overexpression. These data are largely convincing and interesting, but there is a huge gap between these two stories: Why could only these two isoforms of PI3K enhance the axonal transport? What prevents adult CNS neurons from expressing p110 δ isoform? What is the responsible molecular motor for this Rab11/ α 9-integrin (are they comigrating with each other? One possible candidate is KIF13A/B), and what mechanism (motor protein activity or motor-cargo binding) is enhanced by elevated PIP3 level? Is the Rab11 nucleotide cycle (GTP/GDP) modified by p110 δ isoform? Is the transport enhancement resulted from PIP3-mediated nucleotide exchange of Rab11-GDP into Rab11-GTP to facilitate the motor binding? Accordingly, it might be still premature for considering this class of journals and this reviewer would strongly recommend the authors to answer the above questions for improving this manuscript.

Minor points:

- 1) Intensive English text editing would help the readability.
- 2) The Introduction section is especially hard to read. The authors should avoid talking about their own present results but in the last paragraph of the Introduction.
- 3) Figure 1A, C. Lower magnification views should also be included.

Referee #3 (Comments on Novelty/Model System for Author):

In this article, the authors convincingly demonstrate that expression of PI3 kinase (p110) δ in central nervous neurons enhances growth cone PIP3 and stimulates axonal regeneration in the adult CNS. The conclusion fit well with current model of neuronal polarity claiming central role of

PIP3 and PI3K. In essence, the present article suggest that there is a need to restore axonal tip PIP3 for axonal regeneration. Provided the authors are able to answer the following requests, this article could be a landmark in axonal regeneration.

Referee #3 (Remarks for Author):

The results presented here are based on experiments in cultured neurons and in vivo using viral vectors. These data are novel and of high interest in the field of neuronal polarity and axonal regeneration. They were generated using appropriate quantitative methods.

However, two major issues need to be addressed:

1-regarding the medical relevance: the authors need to show functional recovery not just regrowth as in figure 8. This reviewer realizes that this may represent a significant amount of additional work but the authors make strong claims on the medical relevance of their model which then makes this request fully justified.

2-regarding the molecular and cellular mechanism: is there a specific accumulation of the exogenous p110 at the axonal tip? is the effect mediated by Akt, GSK3beta inhibition and CRMP2? what is the effect of inhibiting IGFR1 in cells expressing p110 delta? do the Rab11 endosomes fuse at axonal tips, any evidence for membrane fusion at PIP3 hotspots? could the effect of p110 delta be reversed by pharmacological inhibitors?

Referee #1

This is an interesting manuscript looking at the potential role of PI3K-alpha and -delta isoforms in axonal regeneration. Their investigation pertains to the difference in regenerative capability of the peripheral nervous system compared to the central nervous system. They found that peripheral neurons rely on both PI3Kalpha and delta to generate PtdIns(3,4,5)P3 for axonal regeneration whereas PtdIns(3,4,5)P3 levels are greatly reduced following development in central neurons. Interestingly they found that only expression of p110 δ restored axonal PtdIns(3,4,5)P3 level which increases regenerative axon transport and enhance CNS regeneration in multiple systems and viral expression of p110delta was sufficient to promote regeneration following optic nerve injury. Although the manuscript is generally sound, I see a number of control issues and one general conceptual problem that will need to be addressed either experimentally or at the very least discussed.

Major issues

1- The investigators should use the p110deltaD910A/D910A mouse line and extract neurons which would make this investigation a lot stronger as it would not rely on pharmacology and over-expression only. If the investigators do not have access to this line, at the very least, the authors should use over-expression of p110deltaD910A to assess the effect of blocking PI3K δ activity in all experiments relying on overexpression of PI3Kdelta. Similarly, the specific effects of p110alpha or delta over-expression should be tested using appropriate inhibitor cocktails as in Fig. 1. Finally, no additional effect of inhibitors should be detected upon overexpression of kinase-dead p110deltaD910A.

Regarding the p110deltaD910A/D910A mouse line:

We agree that pharmacological inhibition should ideally be corroborated by additional means, but the p110deltaD910A/D910A kinase inactive mouse has previously been used to confirm that p110 δ functions in DRG neurons to support axon regeneration (Eickholt et al, 2007). In Figure 1, we show that p110 δ inhibition opposes DRG axon regeneration *in vitro*, whilst the paper by Eickholt et al demonstrated sciatic nerve regeneration (DRG axons *in vivo*) was impaired in the p110deltaD910A/D910A kinase inactive mouse. Our findings support the published work, but build on it by investigating the other class 1 PI 3-kinases. We show that both p110 α and δ are required for efficient regeneration, and investigate these throughout the manuscript. The paper by Eickholt et al is now cited in the introduction (top of

page 4) as well as in the results section (near the top of page 6) in the discussion (pages 19, 20 and 22) and the use of the kinase inactive mouse is specifically mentioned on page 20.

Expressing p110 δ ^{D910A} in cortical neurons:

Our subsequent experiments used overexpression of p110 δ to see if we could elevate PIP₃ levels and stimulate regeneration in mature CNS neurons. We originally set out to test hyperactivation of PI3K using p110 δ or the hyperactive p110 α H1047R, and aimed to compare these with wt p110 α and kinase inactive p110 δ ^{D910A}. When we transfected neurons with p110 δ ^{D910A}, using the same expression vectors as for the other constructs, we found that there was a high level of cell death using the kinase inactive construct, and that surviving cells did not look healthy, exhibiting swellings or blebs along neurites. For this reason, we have not used this construct. As class1 PI3Ks are at low levels in mature CNS neurons (Supplementary Figure EV1), it is most likely that overexpressing the kinase inactive construct is interfering with survival pathways normally activated downstream of PI3K activation.

Testing the effects of p110 α or p110 δ overexpression using appropriate inhibitors:

We have performed some of these experiments, and the results are presented in Figure 6 Panel A. We tested the effects of treating neurons overexpressing p110 δ with A66 (p110 α inhibition) or idelalisib (p110 δ inhibition), and found that only idelalisib inhibition prevented the regenerative effects of p110 δ , whilst A66 had no effect. These experiments were then cut short by the Cambridge University closedown. Our last set of experiments would have been to do the same on cells expressing p110 α ^{H1047R} however we were unable to complete these before the closedown. We hope you will agree that the take home message of the paper is that native p110 δ is an effective tool for enhancing regeneration, and that these additional experiments support this. The data are presented in Figure 6A, on a graph together with other inhibitor experiments we performed on neurons overexpressing p110 δ , in response to the issues raised by referee #3.

2- Although type-I PI3K isoforms are expected to act at the level of the plasma membrane, the actual distribution of the delta isoform is in the Golgi. This was shown in macrophages (Low et al., JCB 2010) in cortical neurons and microglia (Low et al., Nat Com 2014). In the Golgi, PI3K δ controls the anterograde trafficking of cytokines such as TNF α (in microglia)

and of APP (in neurons) (Martínez-Mármol et al., J Neurosci., 2019). It is highly surprising that none of these papers are cited or discussed especially considering that some of the results of the manuscript (integrins and Rab11 anterograde transport) actually nicely corroborate these previous reports. Further, the fact that integrins is affected by PI3Kdelta inhibitor was also published previously and not cited in this manuscript (Fiorcari et al., PLoS One 2013). The authors barely mentioned this as a possibility and instead use relatively weak correlative evidence to infer that increased axonal PtdIns(3,4,5)P3 can restore the anterograde transport of integrins as well as the Rab11 in endosomes. It is unclear to me how PI3Kdelta activity promotes these changes at the level of the Golgi and how? This should either be tested or at the very least discussed. Can over-expression of AKT-PH affect such transport? Can AKT-PH bind to the Golgi and colocalize with PI3Kdelta? This could potentially be tested directly on the trafficking of integrins and Rab11.

Regarding p110 δ and the Golgi:

The 2014 paper by Low et al and the paper by Martínez-Mármol were originally referenced in the introduction in the section describing previous neuronal studies on individual PI3K isoforms, however this section was stripped back to include only studies regarding axon growth or regeneration. We have now included the references that are relevant to neurons in the introduction section (page 4), and have rewritten the middle two paragraphs of the introduction to improve readability (also in response to referee #2). We have also cited the 2010 JCB paper, as well as the 2014 and 2019 papers, in the results section relating to figure 6 (the new figure concerning mechanism). This is on page 13. Using the same antibody as was used in the referenced papers, we have observed endogenous p110 δ at what appears to be a Golgi localisation, but we have not used Golgi markers to confirm this. This signal is detected at a low level compared to the overexpressed p110 δ in our mature cortical neuron cultures (DIV 16+ cultures). We do not see overexpressed p110 δ enriched at any particular location, but rather distributed throughout neurons in all locations, only being excluded from the nucleus, and we have included a panel in Figure 6B to show this. We have included a sentence in the results text to say that “Overexpressed p110 δ may therefore exert its effects throughout neurons, including at the Golgi, with the exception of the nucleus” (page 13). We have also cited the paper by Fiorcari et al in the discussion, on page 22.

We have looked for AKT-PH localisation at the Golgi, and also looked for PIP₃ using antibody labelling. Having performed many experiments using cultures from the AKT-PH mice, we do not find any enrichment of AKT-PH at any intracellular locations in either

neuronal or non-neuronal cells, aside from the hotspots we describe in the supplementary figure (Fig EV2). We have also re-examined the many images we have taken of confocal stacks through the cell bodies of mature cortical neurons expressing p110 δ or p110 α ^{H1047R} and labelled for PIP₃, and we do not find enrichment of PIP₃ at intracellular locations.

Regarding the transport of integrins and Rab11:

We initially came to study PI3 kinase through our previous studies on integrins and Rab11. Our hypothesis was that mature axons had insufficient PIP₃ for axon regeneration because PI3 kinase coupled receptors are removed from CNS axons by predominant retrograde transport. We were looking for a way of increasing axonal PIP₃. Our published findings relate to integrins, but we have similar unpublished findings relating to the TrkB receptor, and other labs have found that other growth factor receptors are also not present in abundance in mature CNS axons. Our published findings show that integrins are removed from axons as they mature via a mechanism regulated by activation of the small GTPase ARF6, which also regulates Rab11 transport (Franssen et al 2015, Eva et al 2017, Koseki et al 2017). Because ARF6 activity is strongly regulated by phosphoinositides (via its GEFs and GAPs (Gillingham & Munro, 2007; Hawkins et al, 2006) we reasoned that elevating PI3 kinase activity might be a means of overcoming ARF6 activation. We had performed experiments to look at ARF6 activation downstream of p110 δ but were unsure whether to include the data from these experiments because many pathways will be activated downstream of elevated PI3 kinase activity. Referee #3 has raised issues regarding downstream signalling, and referee #2 has questioned the mechanism of axon transport regulation. We have now included our data that demonstrate regulation of ARF6 activation in neurons expressing p110 δ , in Figure 6 C and D, and this is described in the results text on pages 12 to 15. An additional paragraph has also been added to the methods section on page 29 to describe these methods.

We have now concluded the results section relating to figure 6 by stating that “These transport changes may occur as a result of a reduction in axonal ARF6 activation, although it is also possible that signalling through CRMP2 may also contribute (Rahajeng et al, 2010). Additionally, p110 δ may also be functioning at the Golgi to regulate anterograde transport, as has been shown previously (Martinez-Marmol et al, 2019).”

3- In the same vein, the fact that PtdIns(3,4,5)P3 correlate with the ability of neurons to regenerate is an interesting point of the paper but I feel it could be tested further using PTEN inhibitors. VO-OHPic has previously been used with success to show that PI3Kdelta and

PTEN controls the ratio between PtdIns(3,4,5)P3 and PtdIns(4,5)P2 in neurosecretory cells (Wen et al., Nature Com 2011). It should be used to see if it can increase regeneration in a PI3Kalpha/delta-dependent manner.

We considered using this molecule to inhibit PTEN at the same time as overexpressing p110 δ , to study PIP₃ generation as well as to look at regeneration. However, after much discussion with PTEN experts, we felt there was a question mark over the specificity of this particular inhibitor, and chose instead to silence PTEN using the established shRNA. We found that combining p110 δ expression with PTEN shRNA led to cell death in the majority of cells, with excessive PIP₃ production and odd membrane structures in the surviving cells, preventing these experiments. This is shown in the figure below.

We have also performed optic nerve crush experiments, combining PTEN deletion with overexpression of p110 δ . When examining RGC survival, we found that there was no change in the number of cells surviving compared with single interventions. Regarding regeneration, we found that less axons regenerated at 0.5mm from the crush site (compared to when we expressed p110 δ alone) but that similar amounts of regeneration were observed further into the optic nerve. The data are presented below. The graph on the right is from Fig 7, showing regeneration in the transgenic mice, the middle graph is from Fig 8, showing regeneration after either PTEN deletion or AAVp110 δ overexpression, and the graph on the left is from the combined deletion of PTEN and overexpression of p110 δ .

These data are not included in the manuscript because they are part of a further study aimed at increasing neuronal survival and regeneration in the optic nerve, in which we have combined a number of different strategies aimed at increasing signalling through PI3K. We

think that combined deletion of PTEN and overexpression of p110 δ is perhaps not the best approach to overcoming the problem, as deleting PTEN removes the required negative regulation of the pathway. We are now looking at other ways of controlling PTEN activity, including its regulation by either oxidation, acetylation or phosphorylation.

[Figures for referees not shown.]

Minor points

- *Please use the correct phosphoinositides nomenclature throughout the paper. PtdIns(3,4,5)P3 is spelled in 3 different ways in the paper.*

We have now updated all references throughout the manuscript, referring to *PtdIns(3,4,5)P3* as PIP₃, except when it is first introduced on page 3, when it is referred to as phosphatidylinositol(3,4,5)-trisphosphate, and in the discussion on page 20, when discussing PTEN as a PI(3,4)P₂ phosphatase. We have chosen to refer to *PtdIns(3,4,5)P3* throughout as PIP₃ with the subscript 3, rather than to use the longer *PtdIns(3,4,5)P3*. This is simply to improve the readability of the text.

- *The concentration of inhibitors is correctly used and stated but it would be nice not to have to check in methods to get the info. Concentrations and timing should be included in the legend.*

We have now included the concentrations and timings for the inhibitors in the figure legends for figure one.

- *Because, several concentrations of drugs have been used, it is not clear how much DMSO has been used. It should be stated and the authors should check that there is no effect of DMSO at such concentration (inclusion of a no DMSO condition).*

DMSO volume is now included in the methods section, on page 27. We had extensively confirmed there was no effect from DMSO, with numerous preliminary experiments performed whilst establishing the laser axotomy model of DRG regeneration and inhibition of PI3 kinase. This is now included in the methods section on page 31.

- CAL101/Idelalisib tends to fall out of solution in buffers. How did the author assess the correct concentration?

We have used Idelalisib at a concentration of 500nM. The Okkenhaug lab have extensive experience with this inhibitor, and have never had issues with coming out of solution at this concentration. This concentration is also lower than the plasma concentration reached in patients treated with Idelalisib (Brown et al, 2014), so we think it unlikely that it will fall out of solution at this concentration.

- The authors should harmonize letter font and size across all the figures of the manuscript.

We have been carefully through all of the figures and harmonized letter sizes and fonts.

Referee #2

In this paper the authors presented a very ambitious and interesting idea regarding the difference in axon regeneration capacity between CNS and PNS, from the viewpoint of PI3K biology. First the authors have postulated that PNS-specific expression of p110delta and p110alpha-CA support axonal regeneration and CNS neurons could gain this property by their overexpression. As a molecular mechanism, the authors showed axonal transport of α 9-integrin and Rab11 is enhanced by p110 δ overexpression. These data are largely convincing and interesting, but there is a huge gap between these two stories:

Why could only these two isoforms of PI3K enhance the axonal transport?

What prevents adult CNS neurons from expressing p110 δ isoform?

We have examined the PI3K isoforms that are required for axon regeneration of peripheral nervous system DRG neurons, and found that both p110 δ and p110 α are required (Fig 1). In the CNS, all PI3K isoforms are expressed at low levels, and expression is downregulated at the mRNA level as development progresses (supplementary Fig EV1). We investigated whether we could increase PIP₃ levels and regenerative ability by overexpression of either wt p110 δ or p110 α , and found that wt p110 δ increased PIP₃ levels and regeneration, but that

whilst wt p110 α had no effect, the constitutively active p110 α^{H1047R} could also elevate PIP₃ and enhance regeneration. p110 δ expression increased integrin and Rab11 transport, but we have not examined whether p110 α^{H1047R} can do the same, but would expect so. We have focused on p110 δ for the later part of the paper (Fig 6 to 8) because this is the isoform that we test as an intervention for survival and regeneration in the retina and optic nerve. This is because it behaves like the hyperactive p110 α^{H1047R} whilst in its native state. We have added new data to Figure 6 to show that p110 δ expression increases transport partly by signalling through ARF6, most likely downstream of PIP₃ generation (this is explained in more detail below and on pages 4 and 5 of this document, in response to issues raised by referee #1).

Our data confirm that the PNS has a better regenerative ability than the CNS and suggest that this may partly be because PNS neurons express p110 δ and this is upregulated after a PNS injury (Figure EV1 A). The question of why p110 δ is not expressed in the CNS is a good one. Why is the PNS better at regenerating than the CNS? The PNS has perhaps evolved to be better at regeneration because the periphery is more easily and more regularly injured. Whereas the CNS is protected by the skull and vertebrae, and the severe injuries that penetrate these are often not survivable. We think that p110 δ is expressed where an elevated level of PIP₃ is required, and that the PNS has evolved to use p110 δ to support regeneration. Our findings help to explain the nature of p110 δ , which is most well-known for its role regulating T-cell development, differentiation and function, and this is considered in the discussion on page 19.

What is the responsible molecular motor for this Rab11/a9-integrin (are they comigrating with each other?)

Regarding co-migration, we have previously demonstrated that Rab11 and integrins co-migrate in axons, and that integrins are also transported in ARF6 positive endosomes, (Eva et al, 2012; Eva et al, 2010; Eva et al, 2017; Franssen et al, 2015). ARF6 and Rab11 are both regulators of transport through recycling endosomes, and share effectors (Fielding et al, 2005; Inoue et al, 2008). We discuss the role of ARF6 in integrin and rab11 transport in more detail below.

We think that the most likely molecular motor is Kif5 (Kinesin 1). We have performed detergent free immunisolation of Rab11 or integrin endosomes from differentiated PC12 cells, and analysed the composition of these endosomes by protein mass spectroscopy. The methods for this are in our early ARF6 and Rab11 papers (Eva et al, 2012; Eva et al, 2010),

but the results of our analysis by mass spectroscopy are not published. The motors identified by these studies were Myosin Va (a known Rab11 motor), KIF5 a, b and c, and dynein. Kif5 is the motor protein that is linked to Rab11 endosomes by Protrudin (Matsuzaki et al, 2011; Shirane & Nakayama, 2006) during process outgrowth. We have another manuscript that is currently under review describing CNS regeneration driven by Protrudin overexpression.

One possible candidate is KIF13A/B),

We have not examined a role for KIF13. Both of the isoforms of this motor protein are expressed at very low levels in the cortical neurons we used, both at early developmental time points and in maturity. The table below shows the mRNA expression levels in maturing cortical neurons *in vitro*, compared with KIF5 (see above) which is expressed at much higher levels. This is taken from our published data set, which we also reference in supplementary figure EV1 (Koseki et al, 2017).

	DIV 1	DIV 4	DIV 8	DIV 16	DIV 24
kif5a	43.9	77.5	122.3	131.0	88.1
kif5b	41.8	29.9	23.0	23.1	22.2
kif5c	102.3	88.2	101.8	106.7	122.4
kif13a	3.8	5.1	4.4	3.8	1.6
kif13b	3.0	2.0	1.4	1.1	0.4

and what mechanism (motor protein activity or motor-cargo binding) is enhanced by elevated PIP3 level? Is the Rab11 nucleotide cycle (GTP/GDP) modified by p110 δ isoform?

Is the transport enhancement resulted from PIP3-mediated nucleotide exchange of Rab11-GDP into Rab11-GTP to facilitate the motor binding?

Accordingly, it might be still premature for considering this class of journals and this reviewer would strongly recommend the authors to answer the above questions for improving this manuscript.

We have previously studied the effects on directional axon transport of activating or inactivating Rab11, using the S25N or Q70L constructs. We found that activating with Q70L had little effect, and there was only a small increase in retrograde transport as a result of the S25N construct (Koseki et al, 2017), so we do not think that Rab11 activation state is the switch between anterograde or retrograde transport. There is currently little evidence linking

class 1 PI3 kinase or PI(3,4,5)P with Rab11 activation state, although Rab11 is clearly regulated by other phosphoinositides such as PI(3)P or PI(4)P, and Rab11 activation state is known to regulate intracellular localisation (Campa & Hirsch, 2017). We think the direction of transport is regulated by nucleotide exchange on ARF6, which occurs downstream of GEFs regulated by phosphoinositides, and that Rab11 transport is regulated by its interaction with ARF6 (Eva et al 2017, described in response to referee #1 on pages 4 and 5). We have provided additional evidence regarding the activation state of ARF6, described below, but we have also cited the review by Campa and Hirsch (see above), and mentioned the possibility that Rab11 localisation may also be regulated by activation state or by phosphoinositides (in the discussion on page 21).

Our evidence regarding the activation state of ARF6 is in the new Figure 6 C and D. This new figure focuses on mechanism of action, also in response to issues raised by referees #1 and #3. The accompanying text has been completely rewritten, and is on pages 12-15. Our original text did not explain clearly why we included data regarding integrin and Rab11 axon transport, or how we came to be studying these molecules. We have explained this more clearly in the new text, and also in response to referee #1 on pages 4 and 5 of this document. Briefly, ARF6 is known to regulate directional transport along polarised microtubules, by interacting with the adapter molecules JIP3 and JIP4. This occurs in a complex with the dual Rab11/ARF6 effector Rab11Fip3. This is best shown in the image below from the paper describing the interaction (Montagnac et al, 2009). This paper demonstrates that active ARF6 increases the affinity of JIP3 and JIP4 for dynein, whilst inactive ARF6 allows interaction with Kinesin-1 (KIF5), and that this occurs in a complex with Rab11.

It was this paper that led us to investigate ARF6 in neurons, and our papers on directional regulation of integrin and Rab11 axon transport regulated by ARF6 (Eva et al, 2017; Franssen et al, 2015; Koseki et al, 2017). These papers show that ARF6 activation is governed by two axonal ARF6 GEFs, ARNO and EFA6. These GEFs are regulated by phosphoinositides, as discussed in response to referee #1 above on pages 4 and 5. This is now described in manuscript in the results section starting on page 12.

Accordingly, it might be still premature for considering this class of journals and this reviewer would strongly recommend the authors to answer the above questions for improving this manuscript.

We hope our new data in figure 6 and the new discussion regarding Rab11 address these concerns.

Minor points

1) Intensive English text editing would help the readability.

We have checked the text using Grammarly, and made alterations. These are mostly deletions of unnecessary words. Any other alterations are highlighted in blue.

2) The Introduction section is especially hard to read. The authors should avoid talking about their own present results but in the last paragraph of the Introduction.

We have rewritten the middle two paragraphs of the introduction which we agree were a little cumbersome! We have also made sure that all references to our own work are now in the final paragraph of the introduction.

3) Figure 1A, C. Lower magnification views should also be included.

These were originally included as movie files, but were removed due to limits on supplementary files. After correspondence with the editors, these have now been included as movie files at the magnification they were acquired. These are referred to on page 7 in the results section text as movies EV1-4.

Referee #3

In this article, the authors convincingly demonstrate that expression of PI3 kinase (p110) delta in central nervous neurons enhances growth cone PIP3 and stimulates axonal regeneration in the adult CNS. The conclusion fit well with current model of neuronal

polarity claiming central role of PIP3 and PI3K. In essence, the present article suggest that there is a need to restore axonal tip PIP3 for axonal regeneration. Provided the authors are able to answer the following requests, this article could be a landmark in axonal regeneration.

The results presented here are based on experiments in cultured neurons and in vivo using viral vectors. These data are novel and of high interest in the field of neuronal polarity and axonal regeneration. They were generated using appropriate quantitative methods.

However, two major issues need to be addressed:

1-regarding the medical relevance: the authors need to show functional recovery not just regrowth as in figure 8. This reviewer realizes that this may represent a significant amount of additional work but the authors make strong claims on the medical relevance of their model which then makes this request fully justified.

We accept that expression of p110 δ alone will not translate to a clinical treatment. Establishing recovery of function will be an extremely challenging task. Functional recovery after optic nerve injury remains a highly ambitious goal in mammals, with many hurdles to be overcome such as accurate guidance, target recognition and remyelination. We are currently performing experiments that will test functional recovery after injury in the spinal cord, but we expect these to take between a year and 18 months to complete, as we have found in the past (Garcia-Alias et al, 2009; Garcia-Alias et al, 2011). We are also beginning experiments in the optic nerve allowing long time points for regeneration, but we expect to require further combined interventions for efficient recovery of function, such as improved guidance through the visual pathway. For this we will be studying the additional use of integrins, which we have found to enable guided regeneration in the spinal cord (Cheah et al, 2016).

We did not intend to claim that our approach with viral delivery of PI3K delta is close to achieving this goal, but rather that it targets the PI3K pathway in a “potentially translatable fashion”, by using AAV viral vectors. We stated this because the well-known approach of stimulating regeneration through PTEN deletion (which also targets the PI3K pathway) works best in transgenic animals, and that approach is not translatable. Our AAV mediated delivery approach works better than transgenic animals, (Figs 7 and 8), and AAVs are gaining increasing acceptance for clinical use, particularly in the eye. This is what we meant by a “translatable fashion” in the manuscript.

We would like to address this issue by altering the text of the manuscript. Below is the text from the manuscript in which we make claims regarding clinical relevance. These are at the end of the abstract, and in the last paragraph of the discussion. Towards the end of the abstract we stated “Furthermore, viral delivery of p110 δ promoted robust regeneration after optic nerve injury, demonstrating potential for translational development.” We have now deleted “demonstrating potential for translational development.”

In the last paragraph of discussion (on Page 22) we considered whether AAV PI3K delta could be translationally useful, and mentioned that it would most likely need to be applied together with other interventions:

“The search for translatable methods for promoting regeneration in the damaged CNS continues. A successful strategy will likely involve multiple interventions. Could manipulation of PI3K be translationally useful? PTEN deletion or p110 α ^{H1047R} expression can be oncogenic in dividing cells. However, expression under a neuron-specific promoter targets expression to a non-dividing cell type, and it is unlikely that expression of p110 δ in CNS neurons would lead to cell transformation, particularly given its usual expression in PNS neurons. Our study puts forward AAV-mediated delivery of p110 δ as a novel means of stimulating regeneration through the PI3K pathway in a potentially translatable fashion. We propose that p110 δ should also be considered as an additional intervention with other regenerative strategies that target the PI3K pathway, either through growth factor treatments such as IGF-1 plus osteopontin, (Duan et al, 2015; Liu et al, 2017), expression of activated integrins (Cheah et al, 2016) or pharmacological interventions such as insulin (Agostinone et al, 2018).”

We have now deleted the word translatable from the first sentence, the word translationally (from the third line) and the text “in a potentially translatable fashion”.

2-regarding the molecular and cellular mechanism: is there a specific accumulation of the exogenous p110 at the axonal tip? is the effect mediated by Akt, GSK3beta inhibition and CRMP2? what is the effect of inhibiting IGF1 in cells expressing p110 delta? do the Rab11 endosomes fuse at axonal tips, any evidence for membrane fusion at PIP3 hotspots? could the effect of p110 delta be reversed by pharmacological inhibitors?

is there a specific accumulation of the exogenous p110 at the axonal tip?

We have not detected overexpressed p110 accumulating densely at the axonal tip, but rather we detect a diffuse expression here. This is now included in the new Figure 6 which addresses mechanism (Fig 6B).

is the effect mediated by Akt, GSK3beta inhibition and CRMP2?

We have performed a series of new axotomy experiments to look at pathways downstream of PI3K activity. This has been discussed above in response to referee #1. We had already demonstrated increased phospho S6 in neurons expressing p110 δ or p110 α^{H1047R} , which is reporting on signalling through the Akt to mTOR pathway, but we had not looked at signalling through GSK3beta and CRMP2. We chose to look at signalling through CRMP2 by using the CRMP2 inhibitor lacosamide, as has previously been used to report on CRMP2 in RGC neurons (Leibinger et al, 2019). We found that lacosamide prevented the regenerative effect of p110 δ overexpression, and the results are presented in Figure 6A. We also looked at signalling through mTOR by inhibiting with Rapamycin, and found that Rapamycin also prevented p110 δ dependent regeneration. These experiments confirm that p110 δ stimulates regeneration through two downstream pathways that are both functional downstream of Akt.

what is the effect of inhibiting IGFR1 in cells expressing p110 delta?

We performed two experiments inhibiting IGFR1 with NVP-AEW541, and had inconsistent results, in one experiment p110 δ driven regeneration was almost completely prevented, whilst in the other, we found no effect. We were continuing with further experiments when Cambridge University was closed down due to the coronavirus pandemic. We think it is most likely that p110 δ relies on activation of PI3K coupled receptors such as IGFR1, but our experiments were inconclusive.

do the Rab11 endosomes fuse at axonal tips, any evidence for membrane fusion at PIP3 hotspots?

We have not studied direct fusing of Rab11 with the membrane, but have studied Rab11 localisation, and mostly find it accumulating in the central domain, with some endosomes visiting the growth cone periphery. This is described in our papers from 2010 and 2017 (Eva et al, 2010; Koseki et al, 2017).

could the effect of p110 delta be reversed by pharmacological inhibitors?

Yes, we have addressed this using p110 δ and p110 α inhibitors, and confirmed that the regenerative effects of overexpressed p110 δ are prevented by inhibition of p110 δ with idelalisib, but not by inhibition with the p110 α inhibitor A66. The data are presented in Figure 6A.

References

Campa CC, Hirsch E (2017) Rab11 and phosphoinositides: A synergy of signal transducers in the control of vesicular trafficking. *Adv Biol Regul* 63: 132-139

Cheah M, Andrews MR, Chew DJ, Moloney EB, Verhaagen J, Fassler R, Fawcett JW (2016) Expression of an Activated Integrin Promotes Long-Distance Sensory Axon Regeneration in the Spinal Cord. *The Journal of neuroscience : the official journal of the Society for Neuroscience* 36: 7283-7297

Eickholt BJ, Ahmed AI, Davies M, Papakonstanti EA, Pearce W, Starkey ML, Bilancio A, Need AC, Smith AJ, Hall SM et al (2007) Control of axonal growth and regeneration of sensory neurons by the p110delta PI 3-kinase. *PLoS ONE* 2: e869

Eva R, Crisp S, Marland JR, Norman JC, Kanamarlapudi V, ffrench-Constant C, Fawcett JW (2012) ARF6 directs axon transport and traffic of integrins and regulates axon growth in adult DRG neurons. *The Journal of neuroscience : the official journal of the Society for Neuroscience* 32: 10352-10364

Eva R, Dassie E, Caswell PT, Dick G, ffrench-Constant C, Norman JC, Fawcett JW (2010) Rab11 and its effector Rab coupling protein contribute to the trafficking of beta 1 integrins during axon growth in adult dorsal root ganglion neurons and PC12 cells. *The Journal of neuroscience : the official journal of the Society for Neuroscience* 30: 11654-11669

Eva R, Koseki H, Kanamarlapudi V, Fawcett JW (2017) EFA6 regulates selective polarised transport and axon regeneration from the axon initial segment. *J Cell Sci* 130: 3663-3675

Fielding AB, Schonteich E, Matheson J, Wilson G, Yu X, Hickson GR, Srivastava S, Baldwin SA, Prekeris R, Gould GW (2005) Rab11-FIP3 and FIP4 interact with Arf6 and the exocyst to control membrane traffic in cytokinesis. *EMBO J* 24: 3389-3399

Franssen EH, Zhao RR, Koseki H, Kanamarlapudi V, Hoogenraad CC, Eva R, Fawcett JW (2015) Exclusion of integrins from CNS axons is regulated by Arf6 activation and the AIS. *J Neurosci* 35: 8359-8375

Garcia-Alias G, Barkhuysen S, Buckle M, Fawcett JW (2009) Chondroitinase ABC treatment opens a window of opportunity for task-specific rehabilitation. *Nat Neurosci* 12: 1145-1151

Garcia-Alias G, Petrosyan HA, Schnell L, Horner PJ, Bowers WJ, Mendell LM, Fawcett JW, Arvanian VL (2011) Chondroitinase ABC combined with neurotrophin NT-3 secretion and NR2D expression promotes axonal plasticity and functional recovery in rats with lateral hemisection of the spinal cord. *J Neurosci* 31: 17788-17799

Gillingham AK, Munro S (2007) The small G proteins of the Arf family and their regulators. *Annu Rev Cell Dev Biol* 23: 579-611

Hawkins PT, Anderson KE, Davidson K, Stephens LR (2006) Signalling through Class I PI3Ks in mammalian cells. *Biochemical Society transactions* 34: 647-662

Inoue H, Ha VL, Prekeris R, Randazzo PA (2008) Arf GTPase-activating protein ASAP1 interacts with Rab11 effector FIP3 and regulates pericentrosomal localization of transferrin receptor-positive recycling endosome. *Molecular biology of the cell* 19: 4224-4237

Koseki H, Donegá M, Lam BYH, Petrova V, van Erp S, Yeo GSH, Kwok JCF, ffrench-Constant C, Eva R, Fawcett J (2017) Selective Rab11 transport and the intrinsic regenerative ability of CNS axons. *Elife* 6: e26956

Leibinger M, Hilla AM, Andreadaki A, Fischer D (2019) GSK3-CRMP2 signaling mediates axonal regeneration induced by Pten knockout. *Commun Biol* 2: 318

Matsuzaki F, Shirane M, Matsumoto M, Nakayama KI (2011) Protrudin serves as an adaptor molecule that connects KIF5 and its cargoes in vesicular transport during process formation. *Molecular biology of the cell* 22: 4602-4620

Montagnac G, Sibarita JB, Loubery S, Daviet L, Romao M, Raposo G, Chavrier P (2009) ARF6 Interacts with JIP4 to control a motor switch mechanism regulating endosome traffic in cytokinesis. *Current biology : CB* 19: 184-195

Shirane M, Nakayama KI (2006) Protrudin induces neurite formation by directional membrane trafficking. *Science* 314: 818-821

17th May 2020

Dear Dr. Eva,

Thank you for the submission of your revised manuscript to EMBO Molecular Medicine. We have now received the enclosed reports from the referees that were asked to re-assess it. As you will see the reviewers are now supportive and I am pleased to inform you that we will be able to accept your manuscript pending the following final amendments:

- 1) Please provide a point-by-point response to my editorial comments
- 2) Figures: please provide size bars in the magnifications inserts in the figures 5, 7, 8, EV2, EV3 and EV4.
- 3) Source Data:

We encourage the publication of source data, particularly for electrophoretic gels, blots, but also microscopy images with the aim of making primary data more accessible and transparent to the reader. Would you be willing to provide a PDF file per figure that contains the original, uncropped and unprocessed scans of all or key gels used in the figure (including molecular weight markers)? The PDF files should be labeled with the appropriate figure/panel number (1 file/figure), and should have molecular weight markers; further annotation may be useful but is not essential. The PDF files will be published online with the article as supplementary "Source Data" files. If you have any questions regarding this just contact me.

- 4) In the main manuscript file, please do the following:
 - correct/answer the track changes suggested by our data editors by working from the attached/uploaded document
 - we need only 5 keywords
 - move the Appendix supplementary methods to the main article
 - relabel "Conflict of Interest"
 - in M&M, the statistical paragraph should reflect all information that you have filled in the Authors checklist, especially regarding randomisation, blinding, replication.
 - in M&M, for animal work, confirm that all experiments were performed in accordance with relevant guidelines and regulations. The manuscript must include a statement in the Materials and Methods identifying the institutional and/or licensing committee approving the experiments and the licensing number when appropriate for all animals used. Gender, age, origin of the animals and genetic background must be indicated, along with housing conditions.
 - remove the movies legends from the main text

- 5) Funding:

Please make sure to indicate in our submission system all sources of funding including grant numbers and to whom they are allocated.

- 6) 4) For more information: There is space at the end of each article to list relevant web links for further consultation by our readers. Could you identify some relevant ones and provide such information as well? Some examples are patient associations, relevant databases,

OMIM/proteins/genes links, author's websites, etc...

7) The Paper Explained: EMBO Molecular Medicine articles are accompanied by a summary of the articles to emphasize the major findings in the paper and their medical implications for the non-specialist reader. Please provide a draft summary of your article highlighting

- the medical issue you are addressing, = Problem
- the results obtained = Results
- their clinical impact = Impact

8) Every published paper now includes a 'Synopsis' to further enhance discoverability. Synopses are displayed on the journal webpage and are freely accessible to all readers. They include a short stand first (maximum of 300 characters, including space) as well as 2-5 one sentence bullet points that summarise the paper. Please write the bullet points to summarise the key NEW findings. They should be designed to be complementary to the abstract - i.e. not repeat the same text. We encourage inclusion of key acronyms and quantitative information (maximum of 30 words / bullet point). Please use the passive voice. Please attach these in a separate file or send them by email, we will incorporate them accordingly.

You are also encouraged to suggest a striking image or visual abstract to illustrate your article. If you do please provide a jpeg file 550 px-wide x (250-400)-px high.

9) As part of the EMBO Publications transparent editorial process initiative (see our Editorial at <http://embomolmed.embopress.org/content/2/9/329>), EMBO Molecular Medicine will publish online a Review Process File (RPF) to accompany accepted manuscripts.

In the event of acceptance, this file will be published in conjunction with your paper and will include the anonymous referee reports, your point-by-point response and all pertinent correspondence relating to the manuscript. Let us know whether you agree with the publication of the RPF and as here, if you want to remove or not any figures from it prior to publication.

10) Data and software availability:

To list the primary data generated in your study, we would kindly ask you to include a formal "Data and software availability" section (after Materials & Methods) that follows the example below:

Please use the following format:

- [data type]: [full name of the resource] [accession number/identifier] ([doi or URL or identifiers.org/DATABASE:ACCESSION])

If this doesn't apply to your study, indicates instead: "This study includes no data deposited in external repositories"

I look forward to receiving a new revised version of your manuscript as soon as possible, within 2

weeks if possible.

Yours sincerely,

Celine Carret

Celine Carret, PhD
Senior Editor
EMBO Molecular Medicine

*** Instructions to submit your revised manuscript ***

To submit your manuscript, please follow this link:

<https://embomolmed.msubmit.net/cgi-bin/main.plex>

- 1) a .docx formatted version of the manuscript text (including Figure legends and tables)
- 2) Separate figure files*
- 3) supplemental information as Expanded View and/or Appendix. Please carefully check the authors guidelines for formatting Expanded view and Appendix figures and tables at <https://www.embopress.org/page/journal/17574684/authorguide#expandedview>

***** Reviewer's comments *****

Referee #1 (Remarks for Author):

The authors have tried their best to address my comments in very difficult conditions. I am happy with the revisions and do agree that the current manuscript is highly improved and well worth publishing.

Referee #3 (Remarks for Author):

The authors have satisfactorily answered the reviewers' comments. They have tone down the potential medical impact which remains low at this point.

University of Cambridge,
John Van Geest Centre for Brain Repair,
Robinson Way, Cambridge,
CB2 0PY

Dr Richard Eva: re263@cam.ac.uk

27th May 2020

Celine Carret, Senior Editor, EMBO Molecular Medicine.

Manuscript #EMM-2019-11674-V3: “PI 3-kinase delta enhances axonal PIP3 to support axon regeneration in the adult CNS”.

Dear Dr. Carret,

Thank you for your time with our manuscript. The authors are very pleased that the reviewers were happy with the improved paper, and that you are able to accept our manuscript pending a few changes. We have now addressed the changes you requested and submit our revised manuscript. We have responded to each of the editorial points individually, and these are detailed below. We have also included a visual abstract file and text and for the “Synopsis” and “Paper Explained” sections, and added relevant web links to the end of the manuscript file.

Yours sincerely,

Richard Eva

1) Please provide a point-by-point response to my editorial comments.

Editorial comments are addressed here, and additions or alterations to the manuscript are in purple in the manuscript text.

2) Figures: please provide size bars in the magnification inserts in the figures 5, 7, 8, EV2, EV3 and EV4.

Figure 5 has new scale bars in the insets.

In Figure 7, the insets are at the same magnification, but show the individual colour channels. This is now stated in the figure legends on page 47.

In Figure 8 panel A, the insets are the same magnification and show individual colour channels. This is stated in the figure legends on page 48. The insets in panel F are zoomed images, and now have an additional scale bar.

The inset in figure EV2 now has an additional scale bar.

The inset in figure EV3 Panel A is at the same magnification and shows the individual colour channels, this is now stated in the figure legends on page 50. The insets in figure EV3 Panel B now have additional scale bars.

The insets in figure EV4 now have additional scale bars, except for panel A and C lower panels, which are at the same scale as the main image. This is now mentioned in the figure legends on page 50.

3) Source Data: We encourage the publication of source data, particularly for electrophoretic gels, blots, but also microscopy images with the aim of making primary data more accessible and transparent to the reader. Would you be willing to provide a PDF file per figure that contains the original, uncropped and unprocessed scans of all or key gels used in the figure (including molecular weight markers)? The PDF files should be labeled with the appropriate figure/panel number (1 file/figure), and should have molecular weight markers; further annotation may be useful but is not essential. The PDF files will be published online with the article as supplementary "Source Data" files. If you have any questions regarding this just contact me.

There are no blots presented in the manuscript.

4) In the main manuscript file, please do the following:

- correct/answer the track changes suggested by our data editors by working from the attached/uploaded document.

These changes have been corrected, and the altered/additional text is shown in purple in the figure legends.

- we need only 5 keywords.

There are now only five keywords.

- move the Appendix supplementary methods to the main article.

This has been incorporated into the methods section in the main article, under the "mouse strains" section on page 24.

- relabel "Conflict of Interest" . This has been changed.

- in M&M, the statistical paragraph should reflect all information that you have filled in the Authors checklist, especially regarding randomisation, blinding, replication.

The “Quantitative and Statistical Analysis” section has now been altered to include the information in the Authors checklist, including statements on randomisation, blinding and replication. This is on page 36.

- in M&M, for animal work, confirm that all experiments were performed in accordance with relevant guidelines and regulations. The manuscript must include a statement in the Materials and Methods identifying the institutional and/or licensing committee approving the experiments and the licensing number when appropriate for all animals used. Gender, age, origin of the animals and genetic background must be indicated, along with housing conditions.

We have now included additional information to cover all of these points, at the end of the “Optic nerve injury” section on page 29.

- remove the movies legends from the main text.

These have been removed.

5) Funding: Please make sure to indicate in our submission system all sources of funding including grant numbers and to whom they are allocated.

Funding information is included at submission, as well as in the acknowledgments section on page 23.

6) 4) For more information: There is space at the end of each article to list relevant web links for further consultation by our readers. Could you identify some relevant ones and provide such information as well? Some examples are patient associations, relevant databases, OMIM/proteins/genes links, author's websites, etc...

These have been added to the end of the manuscript on page 51, after the figure legends.

7) The Paper Explained: EMBO Molecular Medicine articles are accompanied by a summary of the articles to emphasize the major findings in the paper and their medical implications for the non-specialist reader. Please provide a draft summary of your article highlighting

- the medical issue you are addressing, = Problem

- the results obtained = Results

- their clinical impact = Impact

This may be edited to ensure that readers understand the significance and context of the research.

Please refer to any of our published articles for an example.

This has been uploaded as a separate document.

8) Every published paper now includes a 'Synopsis' to further enhance discoverability. Synopses are displayed on the journal webpage and are freely accessible to all readers. They include a short stand first (maximum of 300 characters, including space) as well as 2-5 one sentence bullet points that summarise the paper. Please write the bullet points to summarise the key NEW findings. They should be designed to be complementary to the abstract - i.e. not repeat the same text. We encourage inclusion of key acronyms and quantitative information (maximum of 30 words / bullet point). Please use the passive voice. Please attach these in a separate file or send them by email, we will incorporate them accordingly.

This has been uploaded as a separate document.

You are also encouraged to suggest a striking image or visual abstract to illustrate your article. If you do please provide a jpeg file 550 px-wide x (250-400)-px high.

A visual abstract has been uploaded as a JPEG.

9) As part of the EMBO Publications transparent editorial process initiative (see our Editorial at <http://embomolmed.embopress.org/content/2/9/329>), EMBO Molecular Medicine will publish online a Review Process File (RPF) to accompany accepted manuscripts. In the event of acceptance, this file will be published in conjunction with your paper and will include the anonymous referee reports, your point-by-point response and all pertinent correspondence relating to the manuscript. Let us know whether you agree with the publication of the RPF and as here, if you want to remove or not any figures from it prior to publication. Please note that the Authors checklist will be published at the end of the RPF.

We are happy to publish the RPF file, but request to remove the graphs relating to joint PTEN and p110 δ expression in the retina, as these form part of an ongoing study. These are on page 7 of the response to reviewers document that was previously submitted.

10) *Data and software availability:* To list the primary data generated in your study, we would kindly ask you to include a formal "Data and software availability" section (after Materials & Methods) that follows the example below:

Please use the following format: The datasets (and computer code) produced in this study are available in the following databases:

- [data type]: [full name of the resource] [accession number/identifier] ([doi or URL or identifiers.org/DATABASE:ACCESSION])

If this doesn't apply to your study, indicates instead: "This study includes no data deposited in external repositories"

This does not apply to our study, so we have included the statement "This study includes no data deposited in external repositories" in the "Data and software availability" section, after the material and methods.

29th May 2020

Dear Dr. Eva,

Thank you for answering all our questions and attending to all our requests. We are pleased to inform you that your manuscript is accepted for publication and is now being sent to our publisher to be included in the next available issue of EMBO Molecular Medicine.

Please read below for additional IMPORTANT information regarding your article, its publication and the production process.

Congratulations on your interesting work,

Celine Carret

Celine Carret, PhD
Senior Editor
EMBO Molecular Medicine

Follow us on Twitter @EmboMolMed
Sign up for eTOCs at embopress.org/alertsfeeds

*** ** IMPORTANT INFORMATION ** **

SPEED OF PUBLICATION

The journal aims for rapid publication of papers, using using the advance online publication "Early View" to expedite the process: A properly copy-edited and formatted version will be published as "Early View" after the proofs have been corrected. Please help the Editors and publisher avoid delays by providing e-mail address(es), telephone and fax numbers at which author(s) can be contacted.

Should you be planning a Press Release on your article, please get in contact with embomolmed@wiley.com as early as possible, in order to coordinate publication and release dates.

LICENSE AND PAYMENT:

All articles published in EMBO Molecular Medicine are fully open access: immediately and freely available to read, download and share.

EMBO Molecular Medicine charges an article processing charge (APC) to cover the publication costs. You, as the corresponding author for this manuscript, should have already received a quote with the article processing fee separately. Please let us know in case this quote has not been received.

Once your article is at Wiley for editorial production you will receive an email from Wiley's Author Services system, which will ask you to log in and will present you with the publication license form for completion. Within the same system the publication fee can be paid by credit card, an invoice,

pro forma invoice or purchase order can be requested.

Payment of the publication charge and the signed Open Access Agreement form must be received before the article can be published online.

PROOFS

You will receive the proofs by e-mail approximately 2 weeks after all relevant files have been sent to our Production Office. Please return them within 48 hours and if there should be any problems, please contact the production office at embopressproduction@wiley.com.

Please inform us if there is likely to be any difficulty in reaching you at the above address at that time. Failure to meet our deadlines may result in a delay of publication.

All further communications concerning your paper proofs should quote reference number EMM-2019-11674-V3 and be directed to the production office at embopressproduction@wiley.com.

Thank you,

Celine Carret, PhD
Senior Editor
EMBO Molecular Medicine

Corresponding Author Name: Richard Eva
Journal Submitted to: EMBO Molecular Medicine
Manuscript Number: EMM-2019-11674-V3